# Generating metamers of human scene understanding

**Ritik Raina**[1]   **Abe Leite**[1]   **Alexandros Graikos**[1]   **Seoyoung Ahn**[2]   **Dimitris Samaras**[1]
**Gregory J. Zelinsky**[1]

[1]Stony Brook University   [2]UC Berkeley

## Abstract

Human vision combines low-resolution "gist" information from the visual periphery with sparse but high-resolution information from fixated locations to construct a coherent understanding of a visual scene. In this paper, we introduce *MetamerGen*, a tool for generating scenes that are aligned with latent human scene representations. *MetamerGen* is a latent diffusion model that combines peripherally obtained scene gist information with information obtained from scene-viewing fixations to generate image metamers for what humans understand after viewing a scene. Generating images from both high and low resolution (i.e. "foveated") inputs constitutes a novel image-to-image synthesis problem, which we tackle by introducing a dual-stream representation of the foveated scenes consisting of DINOv2 tokens that fuse detailed features from fixated areas with peripherally degraded features capturing scene context. To evaluate the perceptual alignment of *MetamerGen* generated images to latent human scene representations, we conducted a same-different behavioral experiment where participants were asked for a "same" or "different" response between the generated and the original image. With that, we identify scene generations that are indeed *metamers* for the latent scene representations formed by the viewers. *MetamerGen* is a powerful tool for understanding scene understanding. Our proof-of-concept analyses uncovered specific features at multiple levels of visual processing that contributed to human judgments. While it can generate metamers even conditioned on random fixations, we find that high-level semantic alignment most strongly predicts metamerism when the generated scenes are conditioned on viewers' own fixated regions.[1]

## 1 Introduction

Understanding the latent representation of a scene formed by humans after viewing remains a fundamental unanswered challenge in cognitive science (Epstein & Baker, 2019; Bonner & Epstein, 2021; Malcolm et al., 2016; Võ, 2021). What is clear is that humans represent coherent scenes by a mixture of "gist" information encoded from peripheral vision (Potter, 1975; Greene & Oliva, 2009) with high-resolution but sparse information that humans extract during their scene viewing fixations (Larson & Loschky, 2009; Larson et al., 2014; Eberhardt et al., 2016). Related recent work on scene perception has focused on the concept of object and scene *metamers*—generated stimuli that, although physically different from originals, cannot be discriminated as different by humans when viewed under constrained experimental conditions. (Freeman & Simoncelli, 2011; Balas et al., 2009; Rosenholtz et al., 2012). Understanding scene metamerism is important because metamers tell us the level of misalignment between an actual and generated image that is tolerated by humans and judged to be the same. Generated scenes that fail to become metamers also reveal the details that are important to a scene's representation and that, if changed, result in the generation being detected. However, although several paradigms have been used to identify metamers (e.g., same-different tasks, A/B/X tasks, oddity judgment tasks) (Rosenholtz, 2020), this work on scene perception used simple generative models to synthesize textures and shapes that were shown in behavioral experiments to be metameric with what humans perceive in their visual periphery when

---

eye position is fixed. These paradigms were not designed to study how post-gist changes in fixation affect scene metamerism or what objects a person believes to exist in their blurred peripheral view of a scene, which are the problems we engage.

Inspired by these previous studies showing that generated textures and shapes can become metamers for human scene *perception*, we introduce *MetamerGen*, a state-of-the-art generative model that extends the metamer generation approach to human scene *understanding*. Rather than seeking to generate simple patterns that share low-level statistics with peripheral vision, *MetamerGen* better captures a post-gist level of representation reflecting multiple free-viewing fixations. We see this topic as closer to scene understanding, and we consider the scenes generated by *MetamerGen* as hypotheses for what people believe to be in their peripheral vision. In this paper, we will therefore use the term *scene metamer* to refer to two scenes that have an equivalent understanding. Our approach combines a gist-level scene representation extracted from peripherally blurred pixels with higher-resolution and fixation-specific "foveal" representations corresponding to scene-viewing fixations. Scene gist and the objects fixated during viewing are used to generate in the non-fixated blurred pixels a scene context that is aligned with what a human understands to be in their peripheral vision.

We not only show that many of the scenes generated by *MetamerGen* are metamers for human scene understanding, we also model the dynamic evolution of this understanding by leveraging the capability of a latent diffusion model (Rombach et al., 2022) to generate photorealistic images from diverse conditioning signals (Zhang et al., 2023; Ramesh et al., 2022). *MetamerGen* is a latent diffusion model (Stable Diffusion; Rombach et al., 2022), thus we use each viewing fixation as a conditioning signal to obtain an incremental fixation-by-fixation understanding of a scene (Figure 1).

To adapt Stable Diffusion to our task of generating a scene in blurred peripheral pixels, we introduce a dual-stream representation of foveated scenes (i.e., ones with a high-resolution center and blurred periphery) using a self-supervised image encoder (DINOv2) (Caron et al., 2021; Oquab et al., 2024; Darcet et al., 2024). We utilize an adapter-based framework (Mou et al., 2023), where we condition a pre-trained text-to-image diffusion model on fixation-grounded features extracted by DINOv2 feature representations obtained at each of the fixation locations. We complement the fixation representations with the scene context available from peripheral information by adding a second source of conditioning that uses DINOv2 tokens extracted from a blurred version of the same image.

Our conditioning mechanism allows plausible scene hypotheses to be generated from variable input information. We expect more foveal glimpses of a scene will lead to a richer DINOv2 representation that will in turn enable *MetamerGen* to generate increasingly plausible and contextually appropriate content at the non-fixated scene locations, analogous to how human scene understanding becomes more elaborate with more viewing fixations. We see *MetamerGen* as a tool for generating fixation-specific scene understanding hypotheses that cognitive scientists can test in behavioral studies.

We integrated *MetamerGen* into a same-different behavioral paradigm and conducted experiments to identify the generated scenes that are metamers for human scene understanding. In our paradigm, participants viewed a scene for a variable number of fixations (i.e., gaze contingent), followed by a 5-second delay (during which *MetamerGen* generated a scene from the viewing behavior) and then briefly viewed a second scene (200 msec). Their task was to judge whether this second scene was the same or different from the first. We define a scene metamer as a generation that a participant judges to be the same as the real scene that was first viewed. Our post-hoc analysis showed that while all features throughout the visual hierarchy contributed to the understanding of a scene, high-level semantic features emerged as the strongest predictors of scene understanding metamers.

## 2 PRELIMINARIES

### 2.1 IMAGE GENERATION USING LATENT DIFFUSION MODELS

Diffusion models (Sohl-Dickstein et al., 2015; Ho et al., 2020) comprise two opposing processes—a diffusion process that gradually corrupts data and a denoising process that restores information. The diffusion process relies on Gaussian noise of increasing intensity at every step, while the denoising process uses a learned denoiser model to reverse the degradation. By iterating this process, starting from random Gaussian noise, diffusion models generate new samples.

Latent diffusion models (LDMs) (Rombach et al., 2022) reduce the overall cost by applying the diffusion processes in the latent space of a variational autoencoder (VAE) (Kingma & Welling, 2013). Stable Diffusion (Rombach et al., 2022) uses a pre-trained VAE that spatially compresses images $8\times$ with its encoder and decompresses latent diffusion samples with the decoder. The denoiser $\epsilon_\theta(\cdot)$ is a UNet (Ronneberger et al., 2015) with pairs of down and up-sampling blocks, as well as a middle bottleneck block. Each network block consists of ResNet (He et al., 2015), spatial self-attention, and cross-attention layers, with the latter introducing the conditioning information.

The cross-attention layers condition the denoising process by computing relationships between intermediate image features during denoising and a set of given conditioning embeddings, usually text. When $F \in \mathbb{R}^{h \times w \times c}$ represents the intermediate image features during denoising (reshaped to $hw \times c$ for attention computation) and $e \in \mathbb{R}^{n \times d}$ are the $n$ conditioning embeddings, the cross-attention mechanism first projects features into queries and embeddings into keys and values as

$$Q = FW_Q, \; K = eW_K, \; V = eW_V, \quad Q \in \mathbb{R}^{hw \times d_k}, \; K \in \mathbb{R}^{n \times d_k}, \; V \in \mathbb{R}^{n \times d_v} \qquad (1)$$

where $W_Q \in \mathbb{R}^{c \times d_k}$, $W_K \in \mathbb{R}^{d \times d_k}$, and $W_V \in \mathbb{R}^{d \times d_v}$ are learned projection matrices. The cross-attention output is then computed as:

$$\text{CrossAttention}(F, e) = \text{softmax}\left(\frac{QK^T}{\sqrt{d_k}}\right) V \qquad (2)$$

This mechanism allows each spatial location in the image (rows in $Q$) to attend to relevant parts of the conditioning (rows in $K$), with the attention weights determining how much the information in each conditioning embedding contributes to the denoising process at each spatial location.

## 2.2 ADAPTING LATENT DIFFUSION MODELS TO NEW CONDITIONS

In text-to-image LDMs (e.g., Stable Diffusion), cross-attention layers condition image features on text embeddings. An efficient approach for incorporating *additional* conditioning types, without retraining the model from scratch, can be achieved through adapter-based frameworks (Mou et al., 2023). These adapters re-use the learned text conditioning pathways in the LDM to introduce other modalities of conditioning. This is done by introducing trainable components that transform and project new condition signals into a format compatible with the UNet's existing cross-attention mechanisms. This approach has proven particularly effective for incorporating visual conditioning into text-to-image models (Ye et al., 2023; Wang & Shi, 2023; Ye et al., 2025).

## 2.3 SELF-SUPERVISED IMAGE ENCODERS

DINOv2 (Caron et al., 2021; Oquab et al., 2024) is a self-supervised vision transformer trained for hierarchical visual representation learning without manual annotations. Using multiple self-supervised objectives, including a contrastive loss that causes image features that appear together to have similar embeddings and a reconstruction loss that induces patches to redundantly encode information about their surrounding context, DINOv2 represents both local visual details and higher-level semantics. These properties make it an excellent tool to study fixation-by-fixation human scene understanding. Adeli et al. (2023; 2025) showed that self-supervised encoders are capable of capturing object-centric representations without labels and can provide a backbone capable of predicting high-level neural activity in the brain.

## 3 *MetamerGen*: PERCEPTUALLY-INFORMED CONDITIONING

### 3.1 REPRESENTING FOVEAL & PERIPHERAL VISUAL FEATURES

Given an image and a set of fixation locations, such as those fixated by a human during free-viewing, we first aim to extract the *foveal* information from the image at each fixation location and *peripheral* information capturing the overall context. We employ a DINOv2-Base model (with registers) as the feature extractor to obtain these two sources of information. In Appendix Section A.10 we validate our choice of DINOv2 as the feature extractor for *MetamerGen* by showing its superiority to CLIP.

DINOv2 processes $448 \times 448$ images with a patch size of $14 \times 14$, yielding 1024 tokens ($32 \times 32$ grid), each embedded in 768 dimensions (along with a CLS token and four register tokens encoding

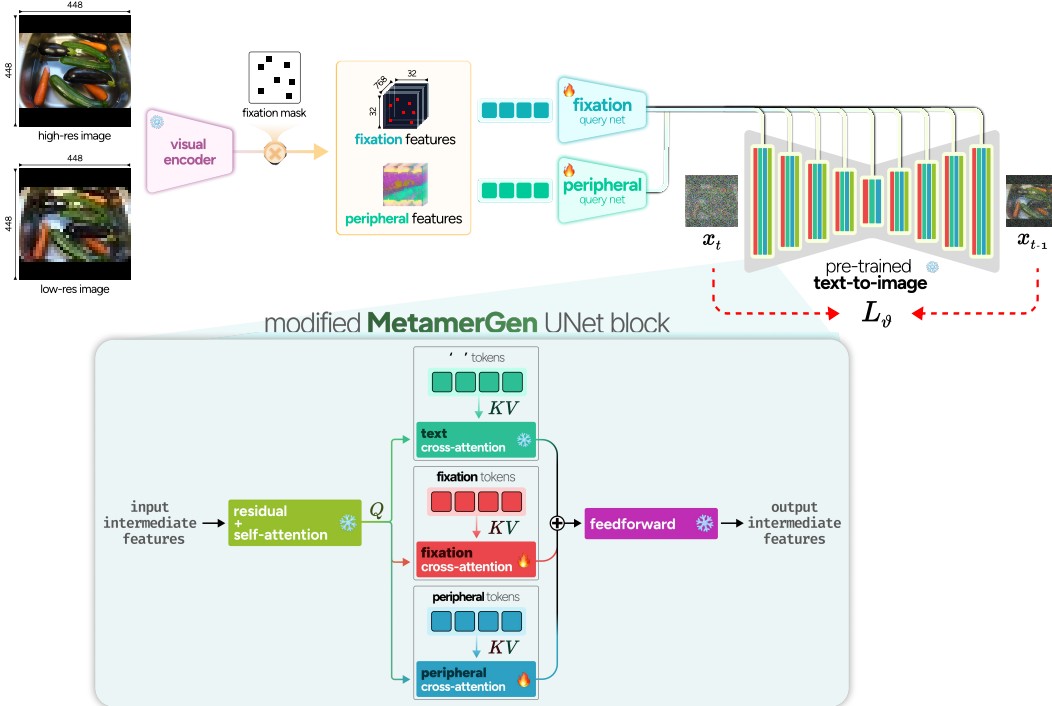

Figure 1: *MetamerGen* **model architecture.** High-resolution and blurred, low-resolution images are processed through DINOv2-Base to extract patch tokens. Foveal features are obtained by applying binary masks to high-resolution patch tokens, retaining only fixated regions. Both foveal and peripheral patch tokens are processed through separate Perceiver-based query networks that compress features into conditioning tokens compatible with Stable Diffusion's cross-attention mechanism. The resulting dual conditioning streams are integrated into the pretrained UNet.

general information about the image). The patch token at a specific location encodes detailed visual and semantic information about that location, analogous to the high-resolution information sampled by the fovea during a fixation. It also encodes local contextual information from around the fixated locations (Adeli et al., 2023; 2025), analogous to the low-resolution parafoveal information extracted by humans. To model the information gathered during a series of fixations, we apply a binary mask $M_{\text{fixation}}$ to the patch tokens extracted from a scene image $I$, corresponding to the image locations fixated by humans, zeroing out all non-fixated image patches.

To obtain peripheral visual features, we downsample the image and then upsample it back to $448 \times 448$. This blurred image, $I_{\text{peripheral}}$, is also processed with DINOv2, but now retaining all output patch tokens without masking. These peripheral tokens encode uncertain visual representations across the entire scene, capturing the noisy information available in peripheral vision that requires validation through targeted foveal fixations (Srikantharajah & Ellard, 2022; Michel & Geisler, 2011).

## 3.2 FOVEAL & PERIPHERAL CONDITIONING ADAPTERS

We develop foveal and peripheral conditioning adapters to integrate visual information as additional conditioning signals in Stable Diffusion. Similar to IP-adapters (Ye et al., 2023), which integrate CLIP image embeddings into Stable Diffusion, we learn how to incorporate DINOv2 patch embeddings into the cross-attention mechanism of the text-to-image Stable Diffusion model.

Both foveal and peripheral DINOv2 embeddings are first processed through separate Perceiver-based resampler networks $R(\cdot)$ (Alayrac et al., 2022; Jaegle et al., 2021) that compress the 1024 DINOv2 embeddings into 32 conditioning tokens compatible with the pre-trained UNet's cross-attention (see Appendix A.3 for more information):

$$e_{\text{foveal}} = R_{\text{foveal}}(\text{DINOv2}(I_{\text{original}}) \odot M_{\text{fixation}}), \ e_{\text{peripheral}} = R_{\text{peripheral}}(\text{DINOv2}(I_{\text{downsample}})) \quad (3)$$

The conditions are then integrated through separate cross-attention mechanisms. For each conditioning source (text, foveal, peripheral) we project separately into keys and values

$$K_c = e_c W_K^c, \; V_c = e_c W_V^c, \quad K_c, \; V_c \in \mathbb{R}^{n_c \times d_k}, \; c = \{\text{text, foveal, peripheral}\} \tag{4}$$

which we then combine additively into the denoising through cross-attention:

$$\text{Attention}(Q, K, V) = \text{softmax}\left(\frac{QK_\text{text}^T}{\sqrt{d_k}}\right) V_\text{text} + \lambda_\text{foveal} \cdot \text{softmax}\left(\frac{QK_\text{foveal}^T}{\sqrt{d_k}}\right) V_\text{foveal}$$
$$+ \lambda_\text{peripheral} \cdot \text{softmax}\left(\frac{QK_\text{peripheral}^T}{\sqrt{d_k}}\right) V_\text{peripheral} \tag{5}$$

$\lambda_\text{foveal}$ and $\lambda_\text{peripheral}$ are scaling factors that control the contribution of either foveal or peripheral visual features to the generation process. In practice we "freeze" the text conditioning, by setting the text caption for all images to an empty string " ".

### 3.3 TRAINING AND INFERENCE

We start from a pre-trained Stable Diffusion 1.5 network (Rombach et al., 2022). The trainable components of *MetamerGen* are the foveal and peripheral resampler networks and their associated key-value projection matrices. Training is conducted on the complete MS-COCO training set (Lin et al., 2015) of approximately $118,000$ images. For foveal conditioning, we apply binary masks that randomly retain $\{1, 2, 3, 5, 10\}$ DINOv2 patch tokens while zeroing all others. This sampling strategy ensures compatibility with our free-viewing behavioral experiments, which constrain scene viewing to a maximum of 10 fixations. At training time, the foveal locations are randomly sampled, whereas at inference time, they are determined by the experimental design. For peripheral conditioning, we blur the images by downsampling to $\{0.0625\times, 0.125\times, 0.25\times, 0.5\times, 1\times\}$ of the original resolution.

To enable robust conditioning in inference, we randomly drop conditions with probabilities $p_\text{foveal} = 0.05$ and $p_\text{peripheral} = 0.10$. The higher peripheral dropout rate prevents over-reliance on peripheral features, which, despite being blurred, retain substantial visual information compared to the sparse foveal features. We employ DDIM (Song et al., 2022) for 50 timesteps, with CFG++ (Chung et al., 2025) and set $\lambda_\text{foveal} = 1.2$ and $\lambda_\text{peripheral} = 0.7$ to balance detail generation with scene plausibility.

We point out that, although *MetamerGen* is conditioned on dense DINOv2 representations of an image (periphery and fixation patch embeddings), the model does not simply reconstruct input images verbatim. We attribute this to the lossiness introduced by the DINOv2 embeddings, as well as stochasticity in the sampling process. We demonstrate this further in Appendices A.7 and A.8.

### 3.4 IMAGE GENERATION QUALITY

We first evaluate the image quality of samples from our model using Fréchet Inception Distance (FID; Heusel et al., 2017) between images generated from *MetamerGen* and COCO-10k-test. Figure 2 shows the results using a single central fixation. Green: We fix the blur level to $0.25\times$, matching the blur level used in our behavioral paradigm, and evaluate how peripheral context affects generation quality by varying the peripheral scale. As peripheral scale increases, FID scores improve as the model better integrates the context coming from the peripheral DINOv2 representations. Red: We evaluated the effect of blur level and found that our model consistently generates plausible scenes for all levels of blur in our evaluation. Blue: We include a text-to-image baseline using SD-1.5 with 10k random captions from the COCO training set.

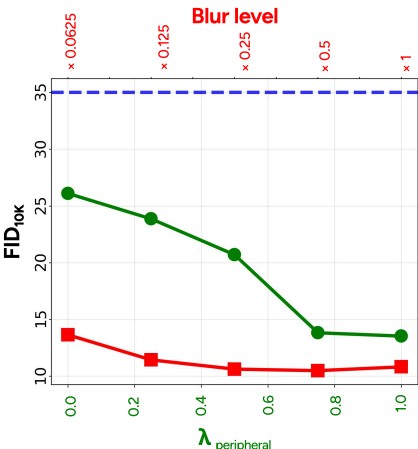

Figure 2: FID values for different input parameters of *MetamerGen*. Lower FID indicates closer alignment with real images and better quality.

*MetamerGen*, fine-tuned on the COCO images, consistently outperforms the text-to-image model, proving that we have successfully integrated images of variable resolution into the conditioning mechanism of Stable Diffusion.

# 4 BEHAVIORALLY-CONDITIONED SCENE METAMERS

## 4.1 PROBING LATENT SCENE REPRESENTATIONS THROUGH SAME/DIFFERENT JUDGMENTS

Metamerism, a term initially established in the color vision literature, has since been applied to questions in texture perception and visual crowding to infer underlying perceptual representations (Freeman & Simoncelli, 2011; Balas et al., 2009; Rosenholtz et al., 2012). In the context of scene understanding, metamers offer a unique opportunity to probe what the visual system encodes and retains after viewing complex natural scenes.

Scene understanding requires the extraction of meaningful structures from complex inputs, including spatial layout, object relations, and global context (Oliva & Torralba, 2006). It is shaped by what the visual system extracts from the input and not the input itself. When a person views scene A and forms internal representation$_A$, then later sees a different scene B and forms representation$_B$, they may believe that these scenes are the same if these representations are aligned. We infer aligned representations when a person responds "same" to a generated version of the scene. These scene metamers therefore reveal what information the brain has encoded and retained from an originally viewed scene, and by exploring the factors causing a generation to become a metamer we are able to investigate the structure of human scene understanding.

## 4.2 REAL-TIME BEHAVIORAL PARADIGM

We developed a real-time same-different behavioral paradigm to evaluate whether *MetamerGen* generates perceptually convincing scene metamers. This paradigm directly tests whether images reconstructed from sparse fixational sampling can achieve perceptual equivalence with the original, thereby revealing the sufficiency of fixated information for scene representation.

**Experimental Design** Participants (n=45) engaged in a paradigm requiring them to first freely view a scene and then make a same-different judgment about that scene. Each of the 300 experimental trials followed the same structured sequence, and eye gaze was tracked throughout each trial (see Appendix A.1 for additional details and a figure illustrating the paradigm). Participants completed a drift check (to maintain eye-data quality) and then freely viewed a natural scene image until reaching a predetermined fixation count $\{1, 2, 3, 5, 10\}$, after which the image was removed. Critically, participants chose their own fixation locations. We systematically varied information availability by manipulating fixation count, testing how additional visual information influenced the generation quality and the rate of metamerism. A 5-second interval followed the offset of the scene during which the participant maintained fixation on an otherwise blank screen. During this interval, *MetamerGen* generated a new version of the original image in real time based on the viewing fixations. Participants were then shown a second image for 200 milliseconds, too brief for an eye movement but sufficient for a perceptual judgment (Broderick et al., 2023; Wallis et al., 2019), and asked to indicate whether this second image was the same or different from the image viewed originally. Although our primary experimental manipulation was whether the second image was the original or a generation from the participant's own fixations, we also had a "random" condition consisting of 12 other participants making a similar same or different judgment, only now between a just-viewed original image and a generation based on a random sampling of fixations in the image.

**Stimulus Selection** Our stimuli were 300 images from the Visual Genome dataset (Krishna et al., 2017), a subset of the YFCC100M dataset (Thomee et al., 2016). We chose this dataset to avoid overlap with the COCO data used in *MetamerGen* training. To maximize visual diversity in our 300 image sample, we used DreamSim (Fu et al., 2023) to cluster Visual Genome images in semantic representational space and selected one representative image per cluster. Images were filtered to exclude challenging elements for current diffusion models (e.g., humans, printed text, clocks).

**"SAME" judgment — metamers**  **"DIFFERENT" judgment**

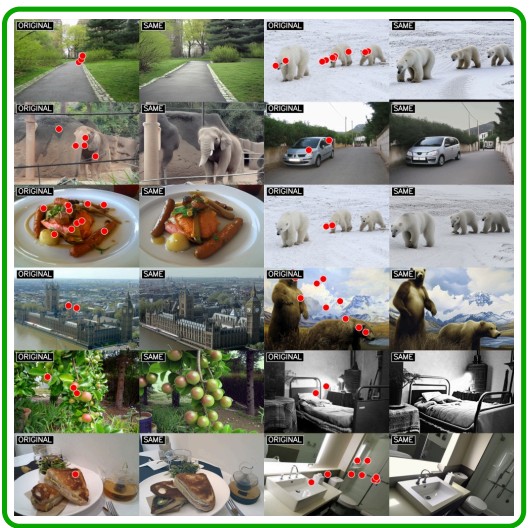 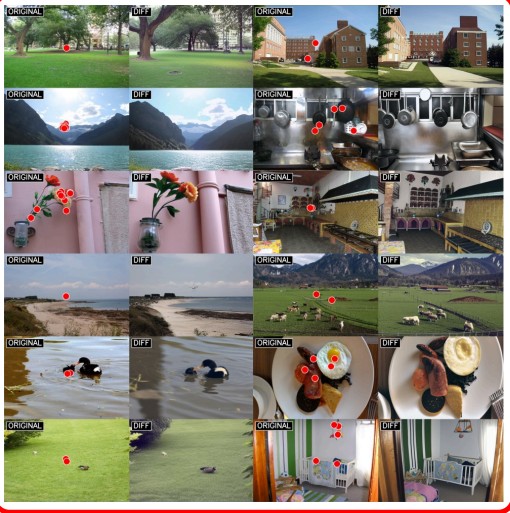

Figure 3: **Metameric vs. non-metameric judgments.** (Left) Original images with human fixations overlaid in red paired with generated images judged as the "same" by participants. (Right) Original images with fixations and generated images judged as "different" by participants. More examples, including generations from random fixations, can be found in Appendix A.6.

## 5 RESULTS: PROBING THE FEATURES OF SCENE METAMERS

As shown in Figure 3, *MetamerGen*'s conditioning on a viewer's own fixations enables it to generate rich and plausible hypotheses for a person's scene understanding. However, from such qualitative results it is difficult to know the features responsible for a generated scene becoming a metamer. We conducted three quantitative analyses to answer this question. First, we compared visual similarity from neurally grounded CNN features with human same–different responses. Our second analysis focused on interpretability by asking which visual features—ranging from low to high levels— shape metameric judgments. Third, we collected new behavioral data to determine the contribution of foveal versus peripheral information by ablating each from the combined model and observing the effect on metamerism rates.

### 5.1 NEURALLY-GROUNDED FEATURE MAPS

To investigate the level of visual processing in the human brain that is most responsible for a scene becoming a metamer, we compared our behavioral judgments to an AlexNet that was trained to be robust to image blur and demonstrated to have internal representations aligned with neural responses from visual brain areas ranging from V1 to inferotemporal cortex (IT) (Jang & Tong, 2024). Using this neurally-grounded model, we isolated contributions from early to late levels of the visual hierarchy to metameric perception. As illustrated in Figure 4, our analysis pipeline treats early, middle, and late layers as proxies for different stages of visual processing. For each layer, we extracted feature maps from both original and generated images and computed cosine similarity to quantify alignment across the visual hierarchy. For attention-guided generations, we found that as feature similarity increased, the proportion of participants judging images as metameric also increased. This relationship held consistently across all layers of the network, from early visual features through high-level representations. These results suggest that metamerism spans the entire hierarchy of visual processing rather than being confined to a single processing stage, and that in order for a scene to be judged a metamer, there must be broad representational alignment across low and high-level visual features.

Although scenes generated from actual fixations and random fixations yielded roughly the same metamer rates (29.4% and 27.7%, respectively, $p = 0.24$), closer comparison hints at an interaction in the mid- and late-layer similarities. For attention-guided metamers, higher similarity to the original predicted more "same" judgments in a consistently linear trend. However, for randomly-guided

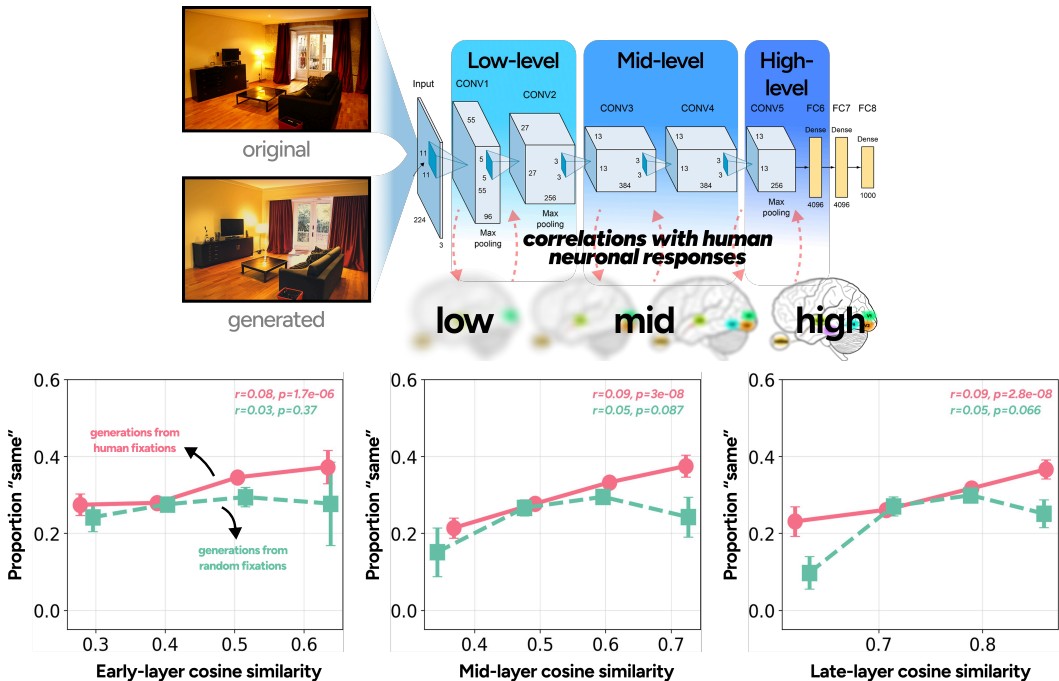

Figure 4: **Multi-level feature analysis pipeline using neurally-grounded model:** (Top) Early, mid, and late network layers serve as proxies for different stages of processing across the hierarchy of visual brain areas. (Bottom) Results show that as feature similarity increased at these different processing levels, the proportion of participants judging generated images as metameric also increased. These effects were clearer when metamers were generated based on a viewer's own fixated locations (salmon) than on randomly-sampled locations (turquoise).

generations, metamerism rates decreased when feature similarity was high. We interpret this trend as suggesting that realistic details in non-fixated regions may expose inconsistencies with the viewer's internal scene representation that make the generated scene discriminable from the original.

## 5.2 INTERPRETABLE VISUAL FEATURE ANALYSIS

Having demonstrated that metameric judgments align with neurally grounded feature similarity, we next performed analyses aimed at making our results more interpretable. We did this by focusing on specific features likely to reflect low-, mid-, and high-level visual processing. We considered a diverse set of features: low-level (e.g., edges, Gabor filters, color), mid-level (e.g., depth cues, proto-object structure), and high-level (e.g., object, semantics, overall perceptual similarity). Because many of these features are correlated, we applied a forward stepwise regression model to identify the most predictive subset ($R^2 = 0.039$), which we focus on in the main text. Detailed contributions of each feature can be found in Appendix A.4.

### 5.2.1 LOW-LEVEL VISUAL FEATURES

We compared human "same" judgments as a function of (i) Gabor filter intensities and (ii) Sobel edge density response differences between the generated and original images to assess how low-level texture features affect similarity judgments. We used normalized responses from Gabor filters oriented at 0°, 45°, 90°, and 135°. Surprisingly, we found that positive differences in Gabor filter responses—where generated images showed stronger texture responses than originals—correlated with more "same" judgments. This suggests that enhanced texture definition, which makes boundaries more distinctive, increases the perceived realism of generated images, even when they differ substantially from the originals (Ho et al., 2012). We also found that greater Sobel edge density responses (Kanopoulos et al., 1988) led to greater "same" judgments, though this effect was redundant with the Gabor filter effect (see A.5).

**mid-level** depth & proto-object segmentation

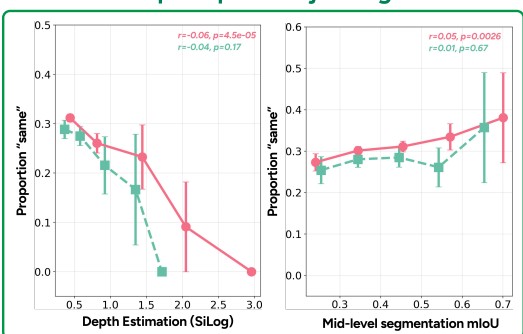

**high-level** visuo-linguistic alignment

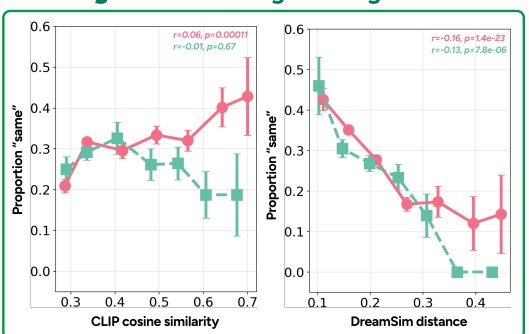

Figure 5: **(Left) Mid-level visual features driving metamer judgments:** For metamers generated from the viewer's own fixation locations (salmon), changes in monocular depth estimates of scene structure strongly predicted "same" judgments to generations. The alignment in proto-object segmentation between original and generated scenes, quantified by mIoU, similarly predicted metamerism rate (higher mIoU scores correlating with higher proportions of "same" judgments), although here the relationship was less pronounced. **(Right) High-level visual features driving metamer judgments:** Semantic similarity strongly predicts metameric scene understanding, with larger DreamSim distances corresponding to reduced perceptual alignment. CLIP similarity shows a similar trend, but not when scenes were generated from randomly-sampled locations (turquoise).

### 5.2.2 MID-LEVEL VISUAL FEATURES

We tested two different mid-level visual features that represent local scene layout information prior to full scene segmentation: (i) relative depth and (ii) proto-object segmentation. We used Depth Anything (Yang et al., 2024a) to obtain depth maps from the original and generated images and compared them using Scale-Invariant Log error (SiLog) (Lee et al., 2021; Eigen et al., 2014). As shown in the leftmost plot from Figure 5, as discrepancies between depth maps increased, the proportion of "same" metameric judgments systematically decreased. This pronounced effect of depth on metamerism rate suggests that a mid-level representation of the depth relationships in a scene, essential for capturing scene layout (Verhoef et al., 2016), is an important factor in determining whether a generated scene becomes a metamer.

To analyze the effect of mid-level grouping on metamer rates, we extracted proto-object segmentations using the `conv3` layer of the blur-trained AlexNet (Jang & Tong, 2024). These mid-layer representations are important to the formation of robust "proto-objects" (Finkel & Sajda, 1992; Rensink, 2000; Yu et al., 2014), which are the pre-semantic building blocks of visual objects created by grouping local features into simple shapes. We found that greater mIoU in proto-object segmentation predicted more "same" judgments (Figure 5), demonstrating that similarity in proto-object segmentation also affects scene metamerism. Of the two mid-level features tested, however, the effect of depth seemed the clearer.

### 5.2.3 HIGH-LEVEL VISUAL FEATURES

To analyze the effect of high-level semantic features on metamerism rate, we used both (i) CLIP (Radford et al., 2021) and (ii) DreamSim (Fu et al., 2023) as learned semantic similarity models. Higher CLIP values indicate greater semantic alignment between original and generated scenes, whereas for DreamSim greater alignment is indicated by lower values.

DreamSim distance provides the clearest evidence for high-level semantic alignment contributing to a metameric judgment. We found that smaller distance values predict more "same" responses to generations (Figure 5, rightmost plot). This is likely because DreamSim was specifically trained on human judgments using a two-alternative forced-choice paradigm to capture human-like notions of visual similarity. CLIP also predicted this relationship between semantic alignment and the likelihood of a generation becoming a metamer (Figure 5). However, metamerism rate increased with semantic alignment only when generations were based on the viewer's own fixations. When generations were from random fixations, higher CLIP similarity did not predict more "same" metameric

judgments. We suggest that this interaction may be due to random fixations often falling on contextually irrelevant regions, thereby exposing semantic details that are not aligned with the viewer's internal scene representation. A similar interaction between own-fixation and random conditions appears in DreamSim, but only at high distances. Together, we interpret our results to mean that scenes generated from a viewer's attention have better high-level semantic alignment with the viewer's internal scene representation. Additional object-level visual feature analyses are discussed in Appendix A.5.

### 5.3 FOVEAL AND PERIPHERAL FEATURES BOTH CONTRIBUTE TO METAMERIC JUDGMENTS

We ran an ablation experiment to isolate the contribution of foveal and peripheral conditioning in *MetamerGen*. We recruited 10 additional viewers to participate in a same-different task similar to the primary experiment reported above, but with four second-image conditions that systematically assessed the impact of conditioning: identical original images (true "same" responses), generated images using both foveal and peripheral conditioning (as in the primary experiment), generated images using peripheral-only, and generated images using foveal-only conditioning. We found that while both foveal and peripheral conditioning played a role in whether a generation becomes a metamer, the role played by peripheral conditioning was greater. As expected, the full model had the highest metamer rate of 54.5%, compared to the second highest rate of 45.8% in the peripheral-only generation condition. The metamer rate in the foveal-only condition was only 8.4%. This much lower rate was due to the generations in that condition correctly capturing fixated details but differing from originals substantially in the non-fixated regions. In contrast, generations in the peripheral-only condition were more likely to be judged metamers because they better captured the global scene structure and layout. Nevertheless, this ablation analysis shows that the additional conditioning from foveal inputs, when combined with the peripheral conditioning, contributes visual and semantic information that leads to generations better aligned with human scene understanding than generations produced from peripheral conditioning alone. We replicated the multi-level visual feature analysis from Section 5.2 for each condition of this ablation experiment in Appendix A.9.

## 6 LIMITATIONS

While *MetamerGen* is effective at generating semantically coherent scenes from sparse visual inputs, it inherits limitations from the pre-trained Stable Diffusion model on which it is built. In our work, we identified two main limitations: (1) difficulties producing fine-grained facial details and accurate limb articulations (Narasimhaswamy et al., 2024; Wang et al., 2025), and (2) generations of text were often unreadable (Yang et al., 2024b) even when directly fixated. To mitigate the effects of these model weaknesses on our behavioral experiment, we excluded images containing such problematic elements. Including these elements would have caused participants to respond "different" due to Stable Diffusion artifacts rather than differences in their own scene representations.

## 7 DISCUSSION

This paper introduces *MetamerGen*, a latent diffusion model for generating images that are aligned with human scene representations enough to become *scene metamers*. *MetamerGen* demonstrates that sparse conditioning based on peripheral gist and fixation-specific details is sufficient to produce human-aligned scene representations, and should be of broad interest to communities ranging from cognitive scientists to machine learning researchers. For cognitive scientists, *MetamerGen* becomes a powerful tool for exploring the human representation of scenes. It enables the testing of fixation-specific hypotheses about what a person's scene understanding would be given their allocation of attention during scene viewing. For machine learning researchers, *MetamerGen* advances generative modeling by leveraging sparsely sampled inputs to produce semantically coherent scene generations that are broadly aligned with human scene representations. This greater human alignment will contribute to next-generation assistive technologies capable of interacting more naturally with humans. Lastly, while we found that attention-guided conditioning yielded stronger correlations across all feature hierarchies (Figs. 4, 5), we also found no significant difference in metamerism rates between random and attention-guided conditions and this has practical implications for the collection of large-scale datasets of scene metamers because it removes the necessity for eye-data collection.

ETHICS STATEMENT

The behavioral data presented in this work was collected in accordance with ethical guidelines determined by the review board at Stony Brook University responsible for overseeing research on human subjects. All subjects provided informed consent before participating in the behavioral experiment in accordance with board-approved protocols and forms. Participation was entirely voluntary, and participants were free to withdraw at any time without penalty. All behavioral data collected for this study, including the gaze-fixation data collected using the EyeLink 1000 eye-tracker, were de-identified (codified) prior to analysis so that there was no link back to the individual participants.

ACKNOWLEDGMENTS

We would like to thank the National Science Foundation for supporting this work through awards 2123920 and 2444540 to GJZ, and the National Institutes of Health through their award R01EY030669, also to GJZ. AL is supported by NSF-GRFP award 2234683. AG is supported by NSF grants IIS-2123920, IIS-2212046 awarded to DS. Any opinions, findings, conclusions, or recommendations expressed in this material are those of the author(s) and do not necessarily reflect the views of the funding agency.

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

# A APPENDIX

## A.1 EXPERIMENT DETAILS

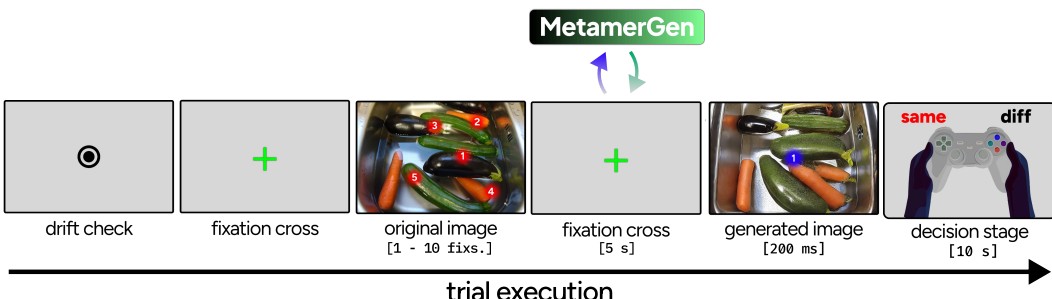

Figure 6: **Real-time paradigm for determining scene metamers.** Each trial begins with drift correction and central fixation, followed by free viewing of an original scene for a predetermined number of fixations. After image offset, participants maintain central fixation for 5 seconds while fixation coordinates are transmitted via API to *MetamerGen* for real-time image generation. The generated image (or original, depending on the condition) is then presented to the viewer for 200ms, followed by an enforced same-different behavioral judgment via a gamepad within a 10-second response window.

For each behavioral trial, we map the fixation coordinate $(x, y)$ of each viewing fixation to the corresponding patch token in DINOv2's $32 \times 32$ grid. For $448 \times 448$ input images, each patch token represents a $14 \times 14$ pixel region (roughly $1.2° \times 1.2°$ visual angle). Fixation coordinates are normalized to this grid space, with the nearest patch token selected and all others zeroed out, forcing the model to reconstruct the entire scene from sparse fixation inputs.

Eye movements were recorded using an EyeLink 1000 eye-tracker (SR Research Ltd., 2006) in tower-mount configuration. This configuration positions the infrared camera above the participant via a mirror, providing an unobstructed view while enabling monocular tracking over a wide visual angle ($55°$ horizontal and $45°$ vertical). The participant, whose head was fixed by a chin rest and head restraint, viewed stimuli on a 27 inch ($2560 \times 1440$ resolution) 240Hz OLED monitor positioned 24 inches from their eyes, which subtended approximately $55° \times 30°$ degrees of visual angle on their retina. Prior to each experimental session, a standard 13-point calibration procedure was performed to ensure accurate eye-gaze tracking. Default algorithms were used to detect eye movements and fixations during both the fixation-dependent free viewing of scenes and all subsequent analyses of eye-gaze fixations.

## A.2 *MetamerGen* TRAINING AND INFERENCE DETAILS

We train following the configuration of Stable Diffusion 1.5 (linear scheduler, fixed variance) for $200K$ steps with a batch size of 32, distributed across 4 NVIDIA H100 GPUs, using the AdamW optimizer with a learning rate of $10^{-4}$ and weight decay of $0.01$. Images from the dataset are padded with 0s to preserve aspect ratios. The model generates output RGB images of size $512 \times 512$.

## A.3 PERCEIVER-BASED RESAMPLER ARCHITECTURE

The Perceiver-based resampler networks $R(\cdot)$ compress variable-length visual embeddings into a fixed number of conditioning tokens suitable for cross-attention in the pre-trained UNet of Stable Diffusion. This architecture is adapted from Alayrac et al. (2022) and Jaegle et al. (2021) Alternative approaches than resamplers like mean pooling or convolutional downsampling would lose spatial relationships and semantic structure in the conditioning tokens (e.g. in our case DINOv2) that are crucial for high-quality image generation.

**Perceiver Attention**   The core component is a cross-attention mechanism that allows a fixed set of learned latent queries to attend to variable-length input sequences (DINOv2 tokens). Given input features $x \in \mathbb{R}^{n \times d}$ and latent queries $\ell \in \mathbb{R}^{m \times d}$, the Perceiver attention computes:

$$Q = \ell W_Q, \quad K, V = \text{concat}(x, \ell)W_{KV} \tag{6}$$

$$\text{PerceiverAttn}(x, \ell) = \text{softmax}\left(\frac{QK^T}{\sqrt{d_k}}\right)V \tag{7}$$

The key insight is that queries come solely from the learned latents $\ell$, while keys and values are computed from both input features $x$ and latents $\ell$ concatenated together. This allows the latents to attend to relevant information in the input sequence while maintaining their learned structure.

**Resampler Architecture**   The full resampler consists of:

- **Learned latents**: $m = 32$ learned query vectors initialized from $\mathcal{N}(0, d^{-0.5})$
- **Input projection**: Linear layer mapping from DINOv2 embedding dimension (1024) to internal dimension $d$
- **Attention layers**: $L = 8$ layers of Perceiver attention followed by feedforward networks with residual connections
- **Output projection**: Final linear projection to match UNet's cross-attention dimension

The resampler processes the 1024 DINOv2 patch embeddings (whether it is via high-resolution fixations or low-resolution peripheral images) and outputs exactly 32 conditioning tokens regardless of input length.

## A.4 Stepwise Regression Model Details

We performed a forward stepwise linear regression analysis to measure the extent to which human judgments could be explained by feature differences in our primary behavioral experiment. The resulting linear model had an $R^2$ value of **0.039**, representing a small but meaningful effect size (in psychological terms). This model incorporated 8 variables, and we evaluated their importance to the model by comparing the full linear model to a model omitting each of them and reporting the change in $R^2$ for each. In descending importance, these variables were: **DreamSim distance** ($\Delta R^2 = 0.10$), **vertical Gabor intensity** ($\Delta R^2 = 0.006$), **predicted depth map RMSE** ($\Delta R^2 = 0.003$), **D3 (Percentage of pixels with depth error $< 1.25^3$ threshold)** ($\Delta R^2 = 0.003$), **mid-level blur-trained CNN feature similarity** ($\Delta R^2 = 0.002$), **CLIP feature similarity** (last hidden layer) ($\Delta R^2 = 0.001$), **CLIP image similarity** (CLS) ($\Delta R^2 = 0.001$), and **D0.25** (Percentage of pixels with depth error $< 1.25^{0.25}$ threshold) ($\Delta R^2 = 0.001$). These results highlight that human scene similarity judgments depend on independent features distributed across the levels of visual processing, and indeed the three most important features in this regression included low-level, mid-level, and high-level measures.

For comparison, we also ran a stepwise regression on the generations conditioned on random fixations. The resulting linear model had an $R^2$ value of **0.031**, meaning that in spite of the generations' variability, we were able to begin explaining scene judgments in this case. However, consistent with our earlier findings that these generations differed from the original image in such unpredictable ways that interpretable predictors were no longer significant, this regression only found 2 significant regressors: **DreamSim distance** ($\Delta R^2 = 0.016$) and **135° Gabor intensity** ($\Delta R^2 = 0.014$).

## A.5 ADDITIONAL DRIVERS OF METAMERIC JUDGMENT

### A.5.1 LOW-LEVEL TEXTURE FEATURES

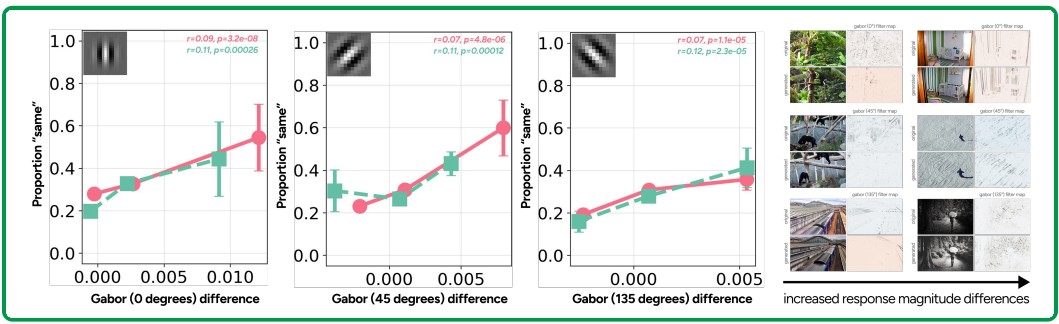

Figure 7: **Stronger Gabor texture responses than originals coincided with greater proportions of metameric judgments.** This suggests that enhanced texture definition, like enhanced edge information, contributes to the perceived realism of generated metamers across multiple spatial frequencies and orientations.

As discussed in Section 5.2, we found that greater oriented edge contrast in the generated image (as measured by Gabor filters) predicts more "same" judgments. Figure 7 presents the results of this analysis visually, demonstrating the effect for 0°, 45°, and 135° filters. (The effect was not significant for the 90°horizontal filter.) The right side of the figure visualizes specific generated images with high or low filter responses at each orientation.

### A.5.2 MULTI-OBJECT RECOGNITION & LOCALIZATION

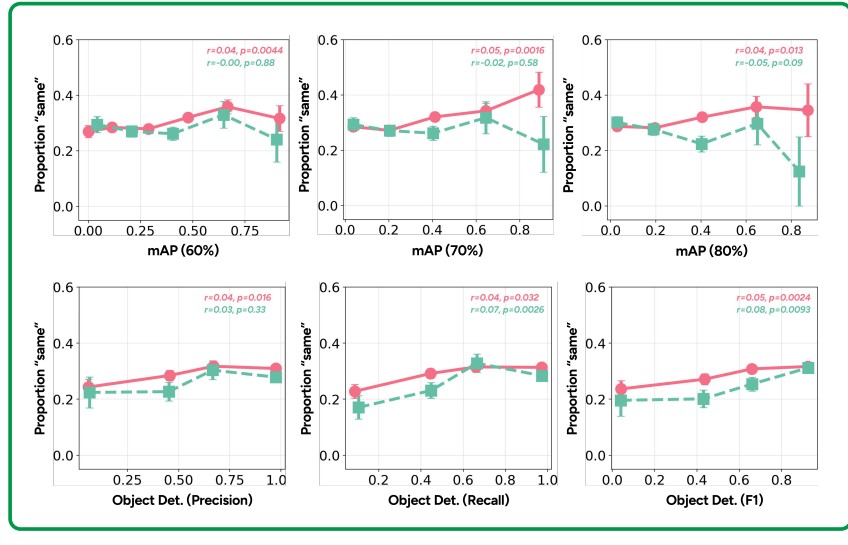

Figure 8: **Object detection errors predict metameric perception:** (Top) mAP scores demonstrate that higher precision accuracies (from mAP 60% to mAP 80%) with better alignment at strict localization boundaries correlate with increased "same" metameric judgments. (Bottom) Object detection metrics show a positive relationship where improvements in model precision, recall, and F1 scores correspond to increased "same" metameric judgments.

To analyze object-level scene understanding, we employed YOLOv8 (Jocher et al., 2023) to extract object detection bounding boxes and class predictions from both original and generated images. Our pipeline compared object inventories between image pairs, quantifying detection errors across multiple metrics: precision (avoiding extra objects), recall (retaining original objects), and localization accuracy measured by mean Average Precision (mAP) at different IoU thresholds.

Analysis revealed that object-level localization inconsistencies systematically impacted metameric perception (Figure 8 (Left)). Localization accuracy showed consistent relationships with metameric perception. As we required increasingly precise object positioning (shown by increasing mAP thresholds), then the gap between human-guided and random fixation conditions systematically widened. This suggests that extremely precise spatial localization becomes increasingly critical for metameric judgments, and that it can best be exemplified using human fixations.

## A.6 ADDITIONAL GENERATION VISUALIZATIONS BASED ON FIXATED INPUTS

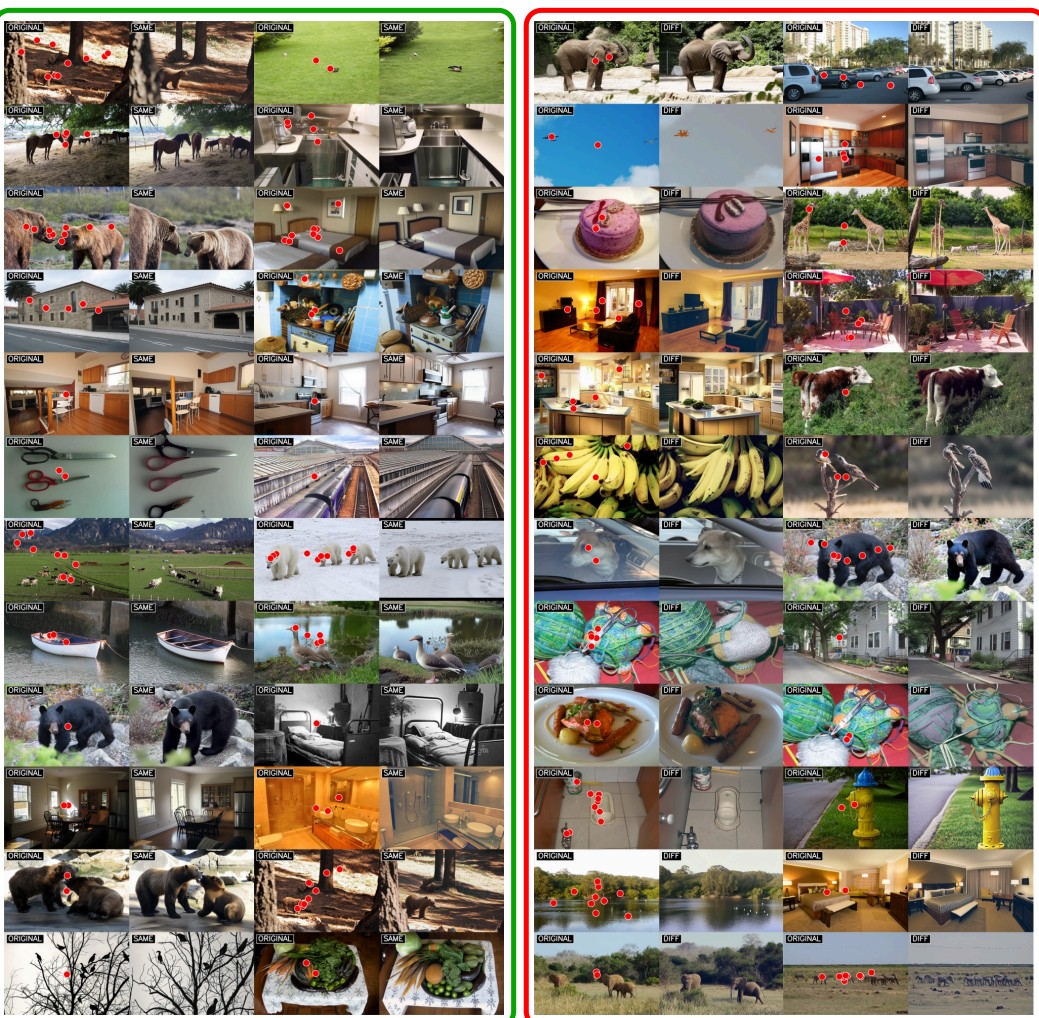

Figure 9: **Additional metameric vs. non-metameric judgment example images based on human fixations.** (Left) Original images with human fixations overlaid in red and corresponding generated images judged as "same" by participants. (Right) Original images with fixations and generated images judged as "different" by participants.

**"SAME" judgment — metamers**       **"DIFFERENT" judgment**

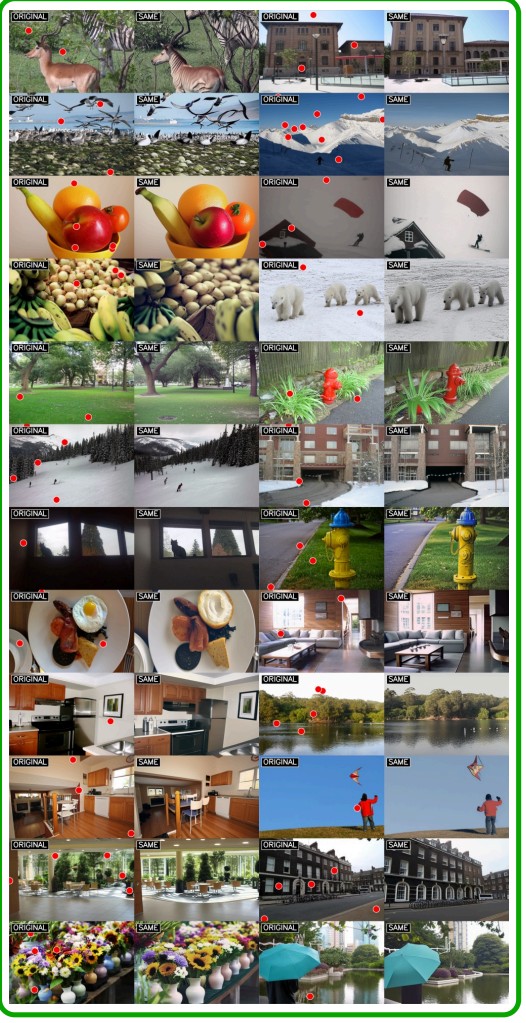 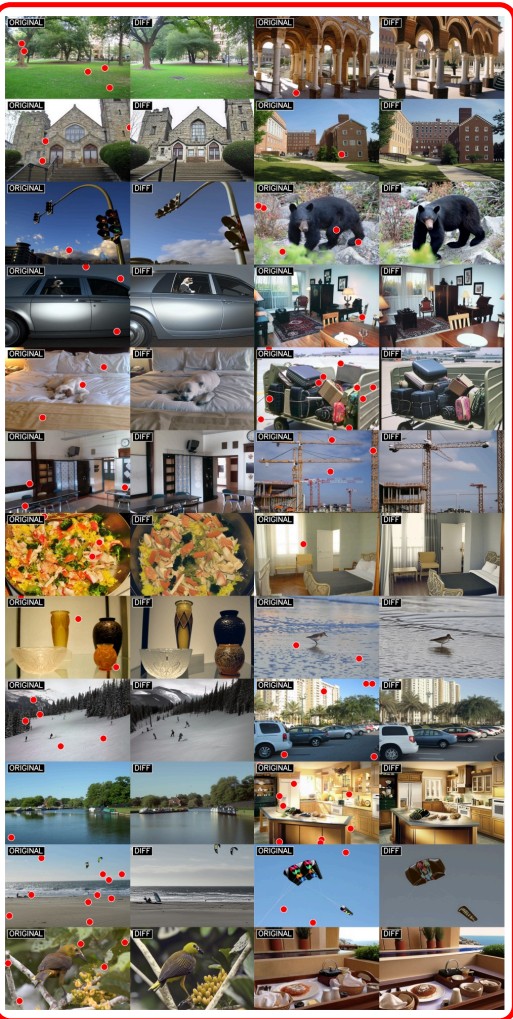

Figure 10: **Additional metameric vs. non-metameric judgment example images based on randomly-sampled fixations.** (Left) Original images with randomly-sampled fixations overlaid in red and corresponding generated images judged as "same" by participants. (Right) Original images with fixations and generated images judged as "different" by participants.

A.7    EFFECT OF PERIPHERAL BLUR AND FOVEAL TOKENS ON IMAGE GENERATION QUALITY

We ran two computational ablation studies to measure how image generation quality on the COCO-10k-test set is affected by (1) peripheral blur level and (2) foveal token count.

For the study of blur levels (Figure 11, Top), we provided peripheral features as the sole input (at varying levels of downsampling) and masked the foveal tokens. We found that greater downsampling blur yields lower CLIP similarities and higher DreamSim distances, as well as higher (worse) FID, though downsampling 0.25x to $112 \times 112$ seems to have little effect on quality compared to keeping the original $448 \times 448$ image, perhaps due to the limited capacity of the DINOv2 embedding and the stochasticity in sampling. Empirically, we find that this introduced just enough uncertainty in the reconstruction that it still resembles the original (CLIP similarity $\sim 0.88$) without maintaining fine details. This is the blur level we choose in our human experiments.

For the study of foveal token count (Figure 11, Bottom), we instead varied the number of randomly-positioned foveal tokens, and provided them as the sole input to *MetamerGen*, masking the peripheral tokens entirely. We found that increasing number of foveal tokens improves CLIP, DreamSim, and FID scores, which nevertheless remain worse than the scores of reconstructions attained by peripheral features only. This highlights the importance of the peripheral conditioning, which we behaviorally confirmed in Section 5.3 and Appendix A.9.

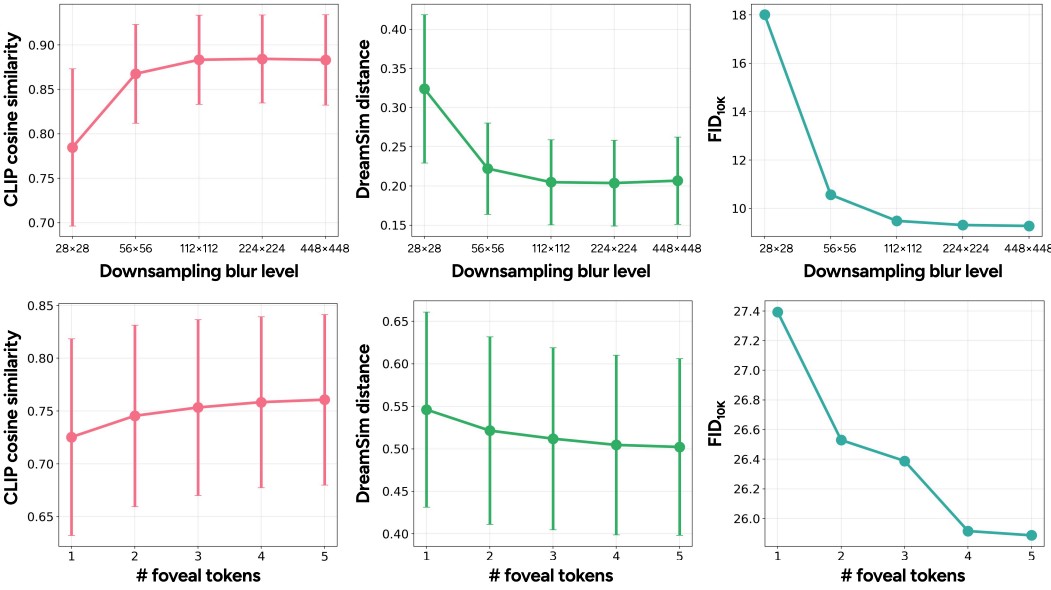

Figure 11: **Influence of blur-level and foveal token count on image generation quality.** (Top Row) Image generation quality decreases as greater blur degrades the base image. (Bottom Row) Image generation quality increases as a function of increasing foveal tokens.

A.8    SEMANTIC SIMILARITY IS MORE IMPORTANT THAN PHYSICAL DISTANCE

While *MetamerGen* was trained to denoise the original image from sparse visual inputs, we found that it never precisely recreated the original images, even when presented with all patch tokens. More importantly, we found that pixel-level similarities between the generated and original images had no effect on whether an image was judged as a metamer. Instead, this judgment was predominantly driven by high-level semantic similarities between the generated and original images.

This result indicates that observers rely primarily on a conceptual and semantic understanding of the scene rather than on low-level pixel features when making metameric judgments. In Figure 12(A), we found that the pixel-level similarities measured by PSNR did not predict whether an image would be a metamer; in Figure 12(B), we establish that in a PSNR–DreamSim plot, low DreamSim distances predict "same" judgments, but high PSNR values do not, with examples of each included.

We conclude that if a researcher wishes to titrate the rate of similarity judgments, they should do so by selecting images based on DreamSim scores, not physical stimulus distance.

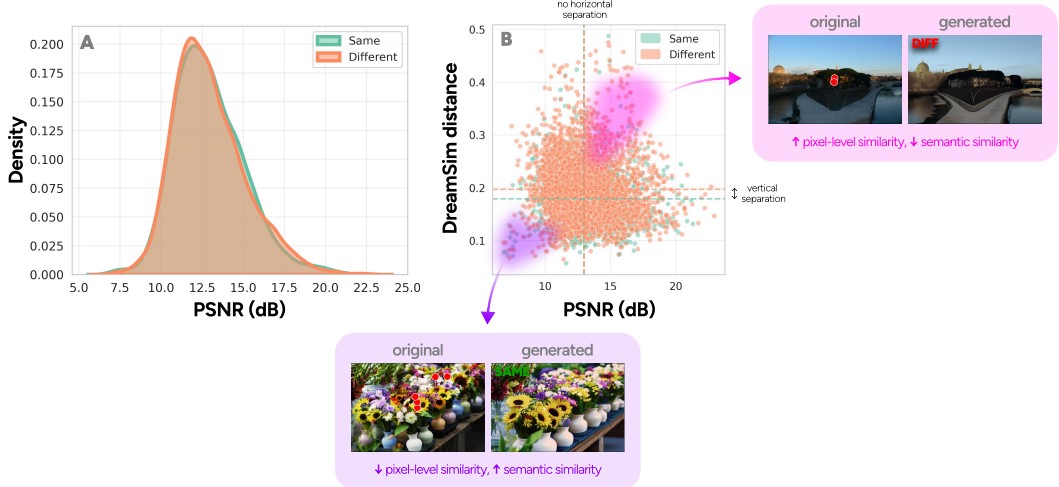

Figure 12: **Comparison of pixel-level (PSNR) and semantic (DreamSim) similarities in metameric judgments.** (A, Left) portrays the histograms of PSNR values for generated images judged as "same" or "different" in the behavioral task, with nearly identical distributions between the two groups. (B, Right) shows the relationship between PSNR and DreamSim distances for all image pairs. There is a clear vertical separation by DreamSim distance that corresponds with metameric judgments, while PSNR values do not discriminate between what is considered metameric.

### A.9 FOVEAL AND PERIPHERAL TOKEN CONTRIBUTIONS TO METAMERIC JUDGMENTS

We present further statistics and analyses of the behavioral ablation experiment presented in Section 5.3. Figure 13 provides qualitative examples of generations from each treatment that were judged "same" and "different", together with a plot of the fool rate in each condition. The generations in the full-model condition are both qualitatively the highest-quality and quantitatively the most likely to fool participants.

One subtlety that emerged during this experiment was that, despite the same participant population and an identical model, participants were substantially more likely to judge images generated by *MetamerGen* as "same" (54.5%) than in the primary experiment (29.4%). Our interpretation of this unexpected pattern of results is that the foveal-only condition, which was generally easy for participants to distinguish from the original image, acted as a low anchor on image similarity, thus lowering participants' threshold for making a same judgment. Because this increases the amount of variability in judgments that can be explained (see below), we see this as a 'feature' rather than a 'bug'.

Figure 14 presents our exhaustive replication of the multi-scale feature correlation analysis from the primary experiment under each condition of the ablation experiment. The main summarizes the two conclusions following from this analysis, which we explain in more detail here.

First, under nearly all metrics, full-model (foveal + peripheral) generations fool participants at a greater rate than peripheral-only generation, even at the same metric scores. This is evidenced by the clear separation between the green and orange lines in most Fig 14 panels. The only features where this gap was not apparent were late-layer cosine similarity and DreamSim distance, suggesting that these metrics may capture a large proportion of the factors that caused humans to judge full-model generations as "same" more than peripheral-only generations.

Second, participant judgments on full-model generations are more explainable than judgments on peripheral-only generations. Along nearly all feature axes (apart from early-layer cosine similarity and mid-level proto-object segmentation mIoU), judgments were more strongly correlated with

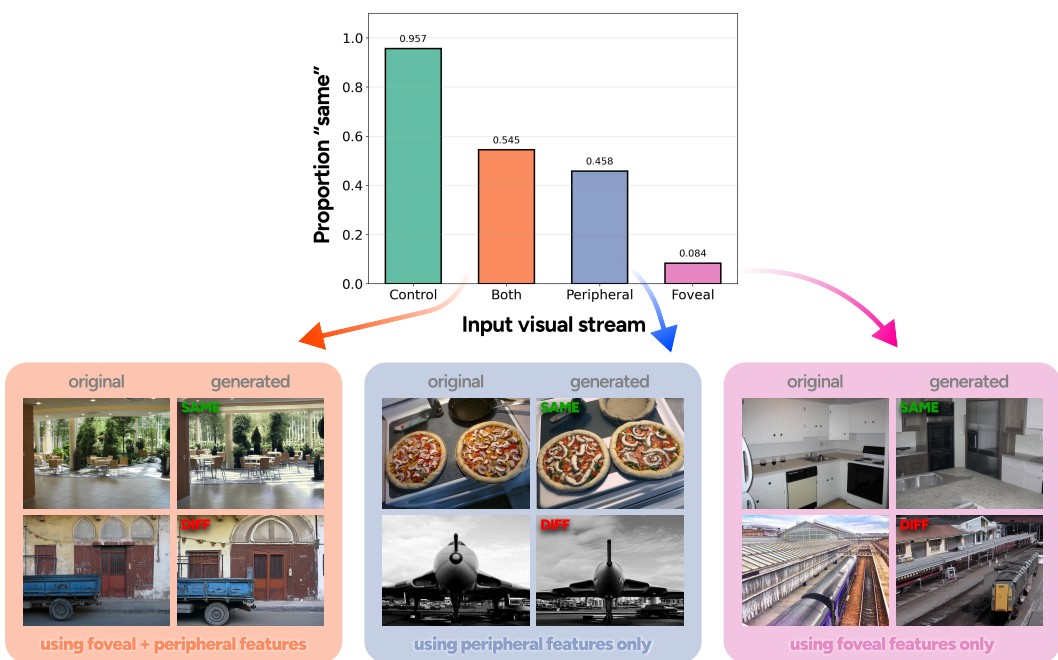

Figure 13: **Higher metameric (same) judgments for images incorporating both peripheral and foveal information.** Using both foveal and peripheral features produced the highest fooling rates. Peripheral-only conditioning yielded the second best results, while foveal-only generations lagged significantly behind. Although the difference between peripheral-only and combined foveal-peripheral conditioning is small, it is meaningful: the additional high-resolution details from fixations lead participants to be more easily fooled.

features under the foveal + peripheral condition. (Under some metrics, they were actually more correlated than they were during the main experiment, thanks to the increased meaningful variance given this set of conditions.) Foveal-only generations were even more poorly explained by these metrics. In fact, the only metric which could explain judgments of foveal-only generations to statistical significance was DreamSim, indicating that these generations, which lacked the gross scene structure and layout provided from the periphery, were so far from the original image that ordinarily important feature axes did not influence judgments.

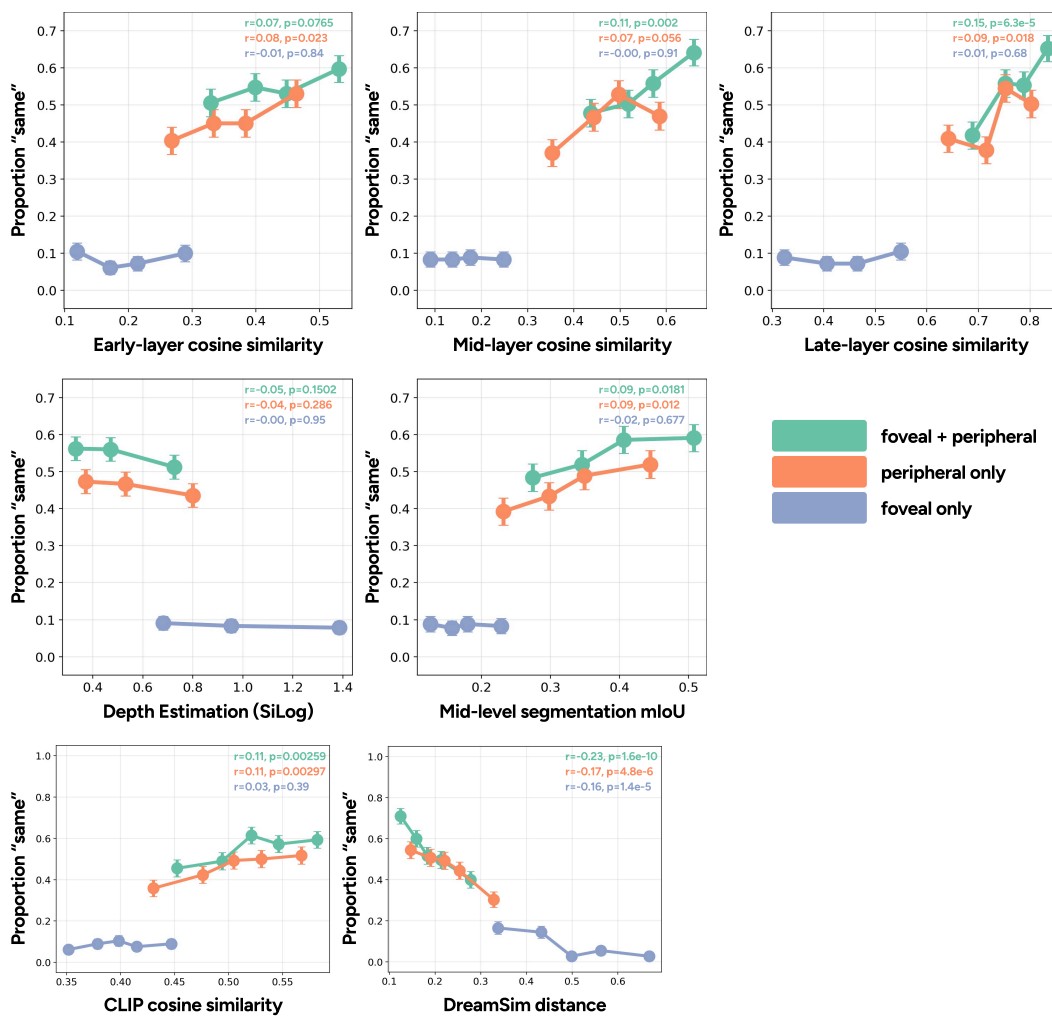

Figure 14: **Impact of input visual streams on hierarchical feature analyses.** (Top Row) Multi-level feature analysis using neurally-grounded model (Jang & Tong, 2024) on driving metameric judgments. (Middle Row) Mid-level visual features driving metameric judgments (mid-level segementention mIoU and SiLog depth estimation). (Bottom Row) High-level visual features driving metameric judgments (CLIP cosine similarity and DreamSim distance).

## A.10 DINOv2 VERSUS CLIP AS THE VISION ENCODER OF *MetamerGen*

Previous adapter-based approaches, like IP-Adapter Ye et al. (2023), have utilized CLIP embeddings as image conditioning inputs for Stable Diffusion. We choose DINOv2 as the visual encoder for foveal and peripheral feature extraction because DINOv2 patch tokens have been shown to better encode both local and contextual information. This contextual encoding emerges from DINOv2's self-supervised training objectives: its reconstruction loss encourages patches to redundantly encode information about their surroundings, while its contrastive loss causes semantically related patches to have similar embeddings (via object and scene structure) (Barsellotti et al., 2025; Adeli et al., 2023). This means a single DINOv2 patch token naturally captures both foveal detail (local information) and parafoveal context (relationships to nearby regions) – precisely the type of representation needed for modeling human fixations.

On the other hand, vision-language models like CLIP optimize for global image-text alignment, which limits their patch-level spatial selectivity and the spatial relationships modeled in their deep layers (Wang et al., 2024; Li et al., 2025). (CNN-based encoders lack emergent representations of patch context – ie, parafoveal information – entirely.)

To empirically validate our choice of DINOv2, we retrained *MetamerGen*, using a CLIP vision encoder, for 100K steps. Using the CLIP-conditioned model, we conducted two image generation ablations using COCO-10K-test images: (1) varying the blur levels for a peripheral-only image generation, and (2) varying the number of foveal tokens for a foveal-only image generation. This ablation is directly comparable to the DINOv2 ablation presented in Appendix A.7.

We observed that CLIP features perform significantly worse at encoding peripheral information, reflected by the higher FID values obtained in Figure 15 (left), and Figure 16. In contrast to DINOv2 tokens, the FID value does not show a significant decrease when we reduce the blur level, meaning that CLIP encodes similar (impoverished) information in its patch tokens for the blurred and non-blurred images, focusing on global scene category, at the expense of more fine-grained scene structure. This is consistent with previous observations made regarding CLIP's relatively low ability to encode contextual information with its patch tokens (Li et al., 2025). For foveal information, increasing the number of tokens has a similar effect for both encoders, though a single DINOv2 token seems to encode more information than a single CLIP token. Overall, our choice of DINOv2 is mainly motivated by its ability to accurately encode both the peripheral and foveal information present in the image.

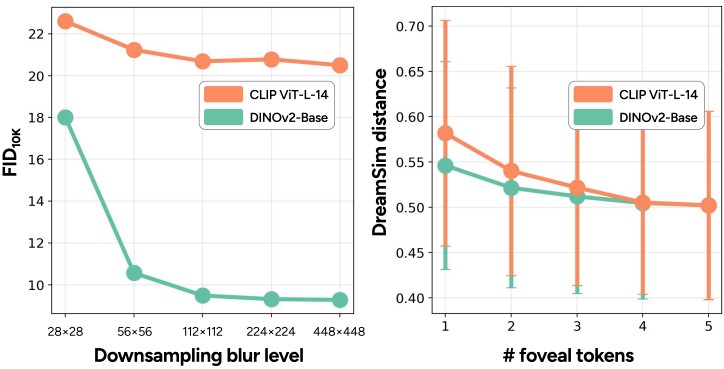

Figure 15: **FID and DreamSim evaluations based on DINOv2 and CLIP as vision encoders for foveal and peripheral feature extraction.** (Left) The image generation quality (FID) for DINOv2-based peripheral generations is consistently better than CLIP patch embeddings. For DINOv2, we observe a sharp drop when decreasing the blur level, showing how decreasing blur results in the model encoding different, more accurate image features. This is not true for CLIP patch tokens, which seem to encode the same limited information across all blur levels. (Right) With increasing numbers of foveal token inputs, the DreamSim distance for both DINOv2 and CLIP-based embeddings decreases. However, DINOv2-based generations yield greater semantic similarities with the original images, especially at low token counts.

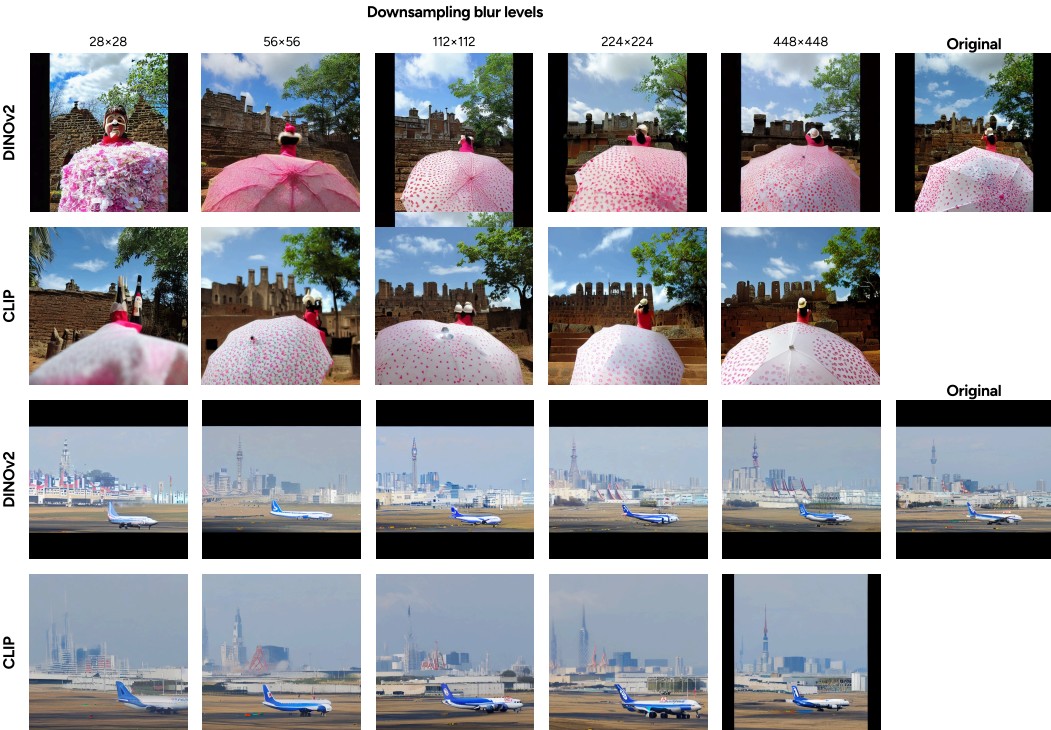

Figure 16: **Image generation examples across blur levels using DINOv2 and CLIP as vision encoders.** DINOv2-based peripheral generations resemble the original images more than CLIP-based generations, even at low blur levels. As the rate of downsampling decreases ($28 \times 28 \rightarrow 448 \times 448$), DINOv2-based generations continue to show substantial improvements while CLIP-based generations exhibit minimal improvements. For the bottom two row pairs, DINOv2-based generations are able to keep the size of the plane (as well as its spatial position) intact irregardless of blur level input. However, that is not the case for the CLIP-based image generations.

