# OpenReview forum: "Generating metamers of human scene understanding"
_ICLR.cc/2026/Conference — ICLR 2026 Oral_

### Official Review · Reviewer_y8G9 · 2025-10-31

**Soundness:** 3
**Presentation:** 3
**Contribution:** 3
**Rating:** 6
**Confidence:** 4

**Summary:**

The paper introduces MetamerGen, an image-to-image framework that synthesizes scene understanding metamers for a viewer’s internal scene representation by conditioning a Stable Diffusion backbone on (i) DINOv2 patch tokens extracted from fixated (high-res) regions and (ii) DINOv2 tokens from a blurred peripheral image. The model is evaluated with a gaze-contingent same-/different behavioral paradigm that measures whether humans judge generated scene maetamers as the “same” scene after a brief re-presentation. The paper further analyzes which levels of visual features (low, mid, high) predict metameric judgments.

**Strengths:**

Formulation of a foveated image-to-image synthesis task that fuses sparse foveal tokens and blurry peripheral tokens into a single latent diffusion generator.

A real-time gaze-contingent same/different behavioral paradigm (45 participants) showing that many generated images are judged “same” (i.e., metameric) and that semantic/high-level alignment best predicts metamerism for human-fixation conditioning.

The paper repurposes the concept of metamers (historically a low-level vision/color concept) as a latent generative modeling problem conditioned on gaze. This is a clever insight that links cognitive science, visual neuroscience, and modern diffusion modeling.
The “foveal + peripheral token fusion” formulation is conceptually neat
Using human fixation maps as structured conditioning signals is a creative way to connect psychophysical data with generative models, and to probe which representations are perceptually sufficient for “scene sameness.”

The method is technically up-to-date and grounded in state-of-the-art visual encoders (DINOv2) and latent diffusion architectures (Stable Diffusion).

**Weaknesses:**

Role / necessity of foveation is not fully ablated or isolated:
The paper reports that MetamerGen can generate metamers conditioned on random fixations as well as human fixations and reports similar overall fooling rates (27.7% vs 29.4%) but then emphasizes the scientific value of human fixations because they produce stronger correlations with feature hierarchies.  This raises two concerns.  If random fixations can produce similar fool rates, how much unique value does foveation / fixation conditioning add for producing perceptually aligned hypotheses (not just for fooling)?
There is no explicit ablation that removes the foveal stream entirely (i.e., peripheral only) or removes the peripheral stream (foveal only) and compares behavioral / feature alignment. A single-factor ablation is missing.

Suitability of DINOv2 patch tokens as an explicit model of foveal or parafoveal signals.
The approach treats a DINOv2 patch token (~14x14 pixels at 448 input → 32x32 grid) as if it corresponds to the information a foveal fixation provides. But DINOv2 was trained on normal (non-gaze-conditioned) images; the authors acknowledge that patch tokens also encode some context. The paper does not show that DINOv2 patch tokens (or their masked/compressed form) are the best representation for foveal information vs alternatives (CLIP token, a CNN trained with foveated inputs, or even raw pixel crops).

Patch ↔ fovea geometric mapping appears simplistic.
A fixation is mapped to the nearest single 14x14 patch token. The visual system’s foveal footprint (receptor sampling + acuity falloff) is more continuous and covers multiple patches; using a single patch may under/overestimate the foveal information.  The authors also need to justify the 14x14 path representing the fovea’s DVA.

Possible confounds in stimuli & filtering
The stimulus set excluded faces/hands etc. to accommodate Stable Diffusions limitations (of course, reasonable), but this restricts generality to many real-world scenes.

Visual vs. Semantic Metamers:
Their psychophysical validation is a same/different task.  Although the paper makes important claims about the semantic nature of the metamers, how are they different from the classic visual metamers?  Would the proposed semantic metamers result in lower ability of observers to tell them apart than using visual metamers?
This seems to be missing in the paper.

**Questions:**

There is a tradition of visual metamers that could be used as a baseline. There is the Freeman and Simoncelli (Nat Neuro) traditional methods, and also newer methods that were developed based on style transfer by Deze at al., 2019 (Neurips).
 In principle, the semantic methods should be better than the pure visual for some tasks (unsure if the one used by the authors).
This should be an important control.

This leads to the next question.  Couldn’t we have two scenes that are clearly different under unlimited viewing, such that observers can easily tell them apart, yet they are still scene metamers because the differences are irrelevant to scene understanding? For example, consider two scenes that are visually distinct but convey the same high-level meaning: a white truck driving on the highway versus a black truck driving on the freeway. Although their low-level image statistics differ, both scenes represent the same event or concept. In that sense, metamerism could arise from equivalence in interpretation, not appearance, suggesting that scene metamers might exist even without any gaze or foveation constraints.


Ablations needed to strengthen the paper:
No-fovea baseline (peripheral-only)
No-periphery baseline (fovea-only)
Tests of DINOv2 suitability

---

> ### Author Response · Authors · 2025-11-23
> **Response to Reviewer y8G9 (Part 1/n)**
>
> We thank Reviewer y8G9 for their careful review of our submission and their comments. Please find our rebuttal to each of the concerns raised below:
>
> ### “Role / necessity of foveation is not fully ablated or isolated”
> We thank the reviewer for bringing up this important point. To comprehensively address this concern, we conducted a new behavioral experiment specifically designed to isolate the individual contributions of the foveal and peripheral conditioning streams in MetamerGen. We recruited 10 additional participants for a full ablation study using our same-different paradigm, where now the second image could be one of the four:
> Identical original images (control)
> Images generated using both foveal and peripheral conditioning (as in the main paper)
> Images generated using only foveal conditioning (participants' fixations without peripheral context)
> Images generated using only peripheral conditioning (peripheral features without fixation features)
>
> This systematic ablation revealed that generated images using both foveal and peripheral conditioning played an important role, substantially increasing fool rate versus the other ablated approaches (peripheral-only and foveal-only). Including both sources of conditioning (foveal and peripheral) leads to the highest fooling rate (54.5%), with the generated images maintaining both the global structures captured by the periphery and the fine details that participants fixated on. The peripheral-only and foveal-rates were 45.8% and 8.4% respectively.
>
> We note that because the model learned to rely on peripheral conditioning for scene structure and foveal conditioning for fine-grained semantic information, images generated using only foveal conditioning tended to be easily visually distinguishable from the original image. This condition acted as a low anchor and appears to have resulted in a more permissive decision threshold for “same” judgments, with participants responding “same” when presented with the original image much more frequently than they did in the main paper experiment. There is a less dramatic but also significant effect where peripheral-only generations fooled participants less often than full-model generations, likely because they lack local semantic conditioning. A Figure showcasing examples from the three settings is added in the Appendix A.9 (Figure 13).
>
> We also performed a correlation analysis of multi-level visual feature differences in this ablation added in Appendix A.9 (Figure 14), which reveals that generated images using both foveal and peripheral conditioning show a higher proportion of “same” responses in comparison with the peripheral-only and foveal-only conditioned images. While the gaps between both (foveal and peripheral) and peripheral-only conditioning trends were small, these differences show how much peripheral contexts matter in scene understanding, as well as the effect that local object-level details that foveal tokens provide. Similar to the results in the main paper, we found the greatest drivers of metameric judgments in this ablation belonged to high-level semantic similarities between original and generated images.
>
> The discussion on the influence of foveal and peripheral features on metameric judgements has been added in Appendix A.9.

---

> ### Author Response · Authors · 2025-11-23
> **Response to Reviewer y8G9 (Part 2/n)**
>
> ### “Suitability of DINOv2 patch tokens as an explicit model of foveal or parafoveal signals”
>
> We thank the reviewer for raising this important question about visual representation choice. A key insight, which both we and the reviewer have already agreed on, is that DINOv2 patch tokens are able to better encode both local and contextual information - where each token contains not only details about its specific 14×14 pixel region but also learned relationships to surrounding patches.
> This contextual encoding emerges from DINOv2's self-supervised training objectives: its reconstruction loss encourages patches to also encode information about their surroundings, while its contrastive loss allows for semantically related patches to have similar embedding information [1,3]. This means that a single DINOv2 patch token can naturally capture both foveal detail (local information) and parafoveal context (relationships to nearby regions) - which matches with the type of representation needed for modeling human fixations. Prior work has demonstrated that DINOv2 patches belonging to the same object exhibit strong feature affinity, creating object-centric representations that mirror how the primate visual system groups features together [3].
> On the other hand, vision-language models like CLIP optimize for global image-text alignment, which limits any patch-level spatial selectivity and relationships by its deep layers [2, 4], and CNNs lack emergent representations of patch context (ie, parafoveal information) entirely.
>
> To empirically validate our choice of DINOv2, we retrained MetamerGen using a CLIP vision encoder for 100K steps. With that, we conducted two image generation ablations using COCO-10K-test images: (1) varying the blur levels for a peripheral-only image generation (2) varying the number of foveal tokens for a foveal-only image generation.
> We observed that CLIP features perform significantly worse at encoding peripheral information, reflected by the higher FID values obtained in Figure 15 (Left), which also do not exhibit a significant decrease when we reduce the blur. This is consistent with previous observations made regarding CLIP's ability to encode contextual information with its patch tokens. For foveal information, increasing the number of tokens has a similar effect for both encoders. Overall, we chose DINOv2 as it is best at encoding both the peripheral and foveal information. We have added this experiment and discussion Appendix A.10.
>
> [1] Barsellotti, L., Bianchi, L., Messina, N., Carrara, F., Cornia, M., Baraldi, L., Falchi, F., & Cucchiara, R., “Talking to DINO: Bridging self-supervised vision backbones with language for open-vocabulary segmentation” ICCV 2025.
>
> [2] Wang, F., Mei, J., & Yuille, A., “SCLIP: Rethinking Self-Attention for Dense Vision-Language Inference” arXiv preprint 2023
>
> [3] Adeli, Hossein, et al. "Affinity-based attention in self-supervised transformers predicts dynamics of object grouping in humans." arXiv preprint 2023.
>
> [4] Li, Yihao, et al., "Does Object Binding Naturally Emerge in Large Pretrained Vision Transformers?." NeurIPS 2025
>
>
> ### “Patch ↔ fovea geometric mapping appears simplistic”
>
> We appreciate the reviewer's attention to the biological plausibility of our mapping. The 14×14 pixel patch size is determined by DINOv2's architecture: it processes 448×448 images into a 32×32 feature grid; thus, these 14x14 patches represent the finest spatial granularity available.
>
> In our experiments, each 14x14 patch was scaled to fill a ~1.2x1.2 dva range on our ~55x30 dva monitor. This comfortably falls between the size of the fovea and the parafovea. While we are actively investigating the best computational representation of foveated information, we deem this to be a wholly satisfactory first approximation.
>
> ### “Possible confounds in stimuli & filtering The stimulus set excluded faces/hands etc.”
>
> We share the reviewer’s concern. Due to Stable Diffusion’s generation artifacts, certain image details, such as faces and hands, would have enough generative defects worthy of being considered “different” in the same-different task. However, we agree with the reviewers' thoughts that removing such real-world details prevents our dataset from covering all perceptual judgments in real-world scenes – particularly considering that there is neural circuitry dedicated to face perception and face similarity judgments. We have added a “Limitations” section in the updated PDF, outlining which images were filtered as well as the reasons mentioned in this response.

---

> ### Author Response · Authors · 2025-11-23
> **Response to Reviewer y8G9 (Part 3/n)**
>
> ### “Visual vs. Semantic Metamers”
>
> Our visual feature analysis between generated and original images was performed to find the level of features within the visual hierarchy that drive metameric “same” judgements. We want to emphasize that, while participants were instructed to respond “different” given any visual difference, no matter how slight, they tended to rely on both low-level visual and high-level semantic cues to reach their decisions. While we found that semantic similarities between the images were the most significant drivers of metameric judgments, this doesn’t mean that low-level feature similarities didn't have any effect entirely. Furthermore, certain low-semantic similar features were still considered “same” in some trials. Classical visual metamers introduced differences in terms of low-level visual features like textures - which may have also been introduced in some of the images that MetamerGen generated.
>
> ### “Visual metamers that could be used as a baseline”
> Traditional visual metamers, like “mongrels”, which are altered images maintaining the same summary statistics as that of the original, are generated and evaluated in the context of a single central fixation. Critically, no existing method can create the mongrel equivalent of a cumulative foveated image; while valuable, such a contribution would rise to the level of its own paper. Generating mongrels in real-time would be both computationally expensive and would also take longer for the image to be shown to the participant during the second-viewing phase. We believe that MetamerGen is capable of producing images (considered as “metamers”) that could be semantically similar, yet different in low-level visual features like textures and color.
>
> ### “Couldn’t we have two scenes that are clearly different under unlimited viewing, such that observers can easily tell them apart, yet they are still scene metamers because the differences are irrelevant to scene understanding?”
> We hope that we understand the reviewer’s question correctly and would be happy to discuss further. As we understand it, we are motivated by a very similar question: which plausible image variations do not rise to the level of “different” judgments? We identified several examples similar to the black-truck / white-truck case outlined by the reviewer, including objects whose details or even subcategories changed. For instance, in a sink full of vegetables, a zucchini became an eggplant in the second viewing, but the participant answered “same”. This example is shown in Figure 9 in the Appendix (last row), along with more examples of “same” and “different” judgements made by participants and their fixations.
> The critical reason why we are studying gaze-contingent scene generation, rather than either unlimited viewing or single-fixation viewing, is that not every viewer reaches the same understanding of a scene: which nouns make it into a viewer’s scene description depends on which objects they fixate (Kollenda et al, 2025). To wit, in the vegetable example, the zucchini was not fixated during initial viewing, so it did not make it into the viewer’s scene understanding.
>
> [5] Kollenda, Diana, Anna-Sophia Reher, and Benjamin de Haas. "Individual gaze predicts individual scene descriptions." Scientific Reports 2025

---

> ### Author Response · Authors · 2025-11-23
> **Response to Reviewer y8G9 (Part 4/n)**
>
> ### “Ablations needed to strengthen the paper”
>
> We present additional ablations for the image modeling part of MetamerGen:
>
> 1. We measure the generated image quality from a no-fovea baseline (peripheral-only) and no-periphery baseline (fovea-only) in Appendix A.8.
> We evaluated the image generation quality of MetamerGen on the COCO-10k-test set when providing peripheral features as the sole input and masking the foveal tokens. We plot the CLIP similarity and Dreamsim distance between the original and generated image, along with the overall FID, when varying the blur-level introduced to extract peripheral features from the base image. We find that when we blur the image to 28x28, down from the original 448x448, we introduce enough uncertainty in the reconstructed image such that, it still resembles the original (CLIP similarity ~0.8) but doesn’t maintain the very fine details.
>
> In the second ablation we varied the number of foveal tokens and provided them as the sole input to MetamerGen, masking the peripheral tokens entirely. In the results we show that increasing the number of foveal tokens improves the generation fidelity, which however still is worse even than using only the peripheral tokens. Results for both ablations are shown in the newly added Figure 11.
> In our human experiments we used both sources of conditioning as they are complementary; the peripheral tokens capture the low-resolution overall structure of the image while the foveal tokens unveil local details in the image. The contribution of each source of conditioning to the human judgements was discussed in “Role / necessity of foveation is not fully ablated or isolated” .
>
> 2. In Appendix A.10., we validate the choice of DINOv2 as the feature extractor by comparing it to CLIP features.
> This is discussed in the reviewer’s question above “Suitability of DINOv2 patch tokens as an explicit model of foveal or parafoveal signals”.

---

### Official Review · Reviewer_KkGc · 2025-11-01

**Soundness:** 4
**Presentation:** 3
**Contribution:** 4
**Rating:** 8
**Confidence:** 4

**Summary:**

The paper is about a very interesting neuroAI study. The problem is to understand when humans consider two pictures to be the same. The authors train a DiT model to generate a different picture (metamer) from an original picture that can fool human subjects into considering they are the same. The DiT is fine-tuned from a pretrained model, adding cross-attention conditioning on the original picture’s features. The features are obtained from DINO. The authors do several in-depth analyses on comparing behavior patterns between different conditions, and between human and CNN and CLIP. The authors also identified that it is the high-level feature that contributes to the metamer perception. Overall, the study provides a novel paradigm to study human scene understanding and shows mechanistic similarity between human scene perception and DNN scene perception.

**Strengths:**

Overall, this is a solid study with rigorous human subject experiments and valid deep learning algorithms. It provides a novel paradigm to study human visual scene perception. The analysis provides insights into the similarity between DNN and human scene perception, and which level of features is critical.

**Weaknesses:**

**Insufficient background on human visual scene perception**

Considering the paper is submitted to a machine learning venue, neuroscience/psychology jargon should be explained. Terms like "metamer" should be defined clearly early on, and traditional study paradigms on them should be introduced.

**Figure 1**

Inconsistent color in Figure 1. in the top part, the fixation is colored blue, but in the bottom part, it is red.

**Structure**

Important results, such as line 478-479 "While MetamerGen was trained to predict images from randomly sampled locations—and
such generations fooled participants at a rate comparable to those conditioned on their own fixations
(27.7% vs. 29.4%)" is hidden in the discussion, rather than in the result section. This is important information for knowing how effective the generated metamers are under different conditions.

**Questions:**

**Physical distance between metamer and original image**

The training and inference process does not regularize the distance between the generated image and the original image. Is it possible their training process learn to recover the original image based on fixation and peripheral condition. So, the generated images are for sure metamers because they are the same or very similar to the original image? Given that the generated images are not always metamers, is there a way to regularize, such that by controlling the distance in some metric space, the generated images are more likely to be metamers?

---

> ### Author Response · Authors · 2025-11-23
> **Response to Reviewer KkGc**
>
> We thank Reviewer KkGc for their time to review our work and their thoughtful comments. Please find our responses to each of your concerns below:
>
> ### Insufficient background on human visual scene perception
>
> We have added more information in the introduction to define metamerism, explain why metamers are valuable for probing human scene understanding, and describe the traditional behavioral paradigms (e.g., same-different tasks) used to study metameric perception in cognitive science. We hope to reach an interdisciplinary audience with this paper; please let us know if any terms remain unclear from the machine learning perspective and we would be happy to add those as well.
>
> ### “Figure 1”
>
> We have fixed the fixation feature colors in the top part of Figure 1 in the updated PDF submission. We also spotted an error in Figure 2, where the top label ticks were incorrectly reversed. This has also been fixed.
>
> ### “Structure”
>
> The fool rate comparisons between random and human fixation conditions (27.7% vs. 29.4% fooling rates) has been moved from Discussion (lines 478-479) to Results (Section 6), where it fits better. We have retained the interpretive discussion of these findings in the Discussion section.
>
> ### “Physical distance between metamer and original image”
>
> We note that our training set (COCO) had no overlap with our experimental set (a subset of Visual Genome/YFCC-300m); however, it is possible that the Stable Diffusion training set did overlap with YFCC-300M. To address whether our model simply learns to recover the original image, we analyzed both pixel-level (PSNR) and semantic-level (DreamSim) similarities between MetamerGen image generations and original images. We found that PSNR results showed no difference between image pairs judged as metamers versus distinct (p=0.83, Cohen's d=0.006), while DreamSim strongly separated these conditions (p<0.05, d=0.36). This demonstrates that metamer judgments are driven by high-level semantic features rather than just simple image reconstruction. We provide these additional detailed results in Appendix A.8 and the added Figure 12.
>
> Furthermore, our model is not able to fully reconstruct a given image with a limited number of fixations, such as the <10 fixations participants make during our human experiment. The diffusion process is conditioned on the **lossy** peripheral features, obtained by blurring the image, and a small set of high-resolution fixation features, obtained from the fixation locations. In Figure 11 in the Appendix, we show that despite the generated image being closer to the reference (in terms of perceptual similarity) as we increase the foveal tokens, there is still a noticeable difference between the two.

---

### Official Review · Reviewer_jZsD · 2025-11-01

**Soundness:** 3
**Presentation:** 3
**Contribution:** 3
**Rating:** 6
**Confidence:** 3

**Summary:**

This paper introduces MetamerGen, a computational model designed to generate visual "metamers"—images that are physically different from an original but are perceived as the same by a human observer. The model, based on a latent diffusion architecture, is conditioned using a dual-stream input that mimics human vision: sparse, high-resolution features from specific fixation points (foveal vision) and blurred, low-resolution features from the entire scene (peripheral vision). This paper utilized a latent diffusion transformer to reveal how humans perceive visual information.

To validate the model, the authors conducted a real-time behavioral experiment where a participant's eye movements were tracked while viewing an image. These fixation data were then used to generate a new image with MetamerGen, which the participant had to judge as "same" or "different" from the original. The study's analysis reveals that high-level semantic similarity between the original and the generated image is the strongest predictor for a "same" judgment.

**Strengths:**

This paper introduced a method to understand how humans process visual information, and what kind of information/feature matters in visual perception.

This paper designs a reasonable experimental setting to interpret human scene understanding by tracking eye movement and leveraging the movement to generate "fake" images using LDM, and asks the tester to justify the fake images.

The findings provide strong evidence that human memory and understanding of a scene rely heavily on high-level semantic information (e.g., object identity and spatial relationships) rather than on precise, pixel-level details.

**Weaknesses:**

This paper only considers a fixed-resolution scenario. It would be interesting to investigate whether varying aspect ratio has an impact on human scene understanding. For example, give a larger resolution version of the original image or give a 9:16 image (the original is 1:1), which adds or removes some side content and maintains the center content.

**Questions:**

N/A

---

> ### Author Response · Authors · 2025-11-23
> **Response to Reviewer jZsD**
>
> We thank Reviewer jZsD for their careful review of our submission. Please find our rebuttal to your critiques below:
>
> ### This paper only considers a fixed-resolution scenario
>
> We chose the 9:16 aspect ratio for our monitor setup to align with the full trackable range of our eye-tracker. Furthermore, the current standard in eye-tracking research predominantly uses 9:16 monitors in experimental designs, which influenced our decision to adopt this fixed-resolution approach. We acknowledge that investigating the impact of different aspect ratios on human scene understanding is indeed an interesting question. However, we cannot conduct this experiment at present since all monitors in our lab are at 9:16. While this limitation prevents us from addressing this suggestion immediately, we would be willing to explore this question with further commitment of resources in the future.
>
> Another way to explore non-fixed resolutions would be to pan and zoom the stimulus differently during the first and second presentations, but that is out of the scope of our paper, which defines metamers as visually indistinguishable stimuli. Further, multiple pan/zoom views are infeasible given the limited output resolution of SD 1.5 (512x512) which we use as the base of our MetamerGen model.

---

### Author Response · Authors · 2025-11-23
**Response to all reviewers**

We thank all the reviewers for their feedback and valuable suggestions. Based on all the comments made, we have revised our submission PDF accordingly.

In our updated submission:
1. We have introduced the concept of “metamerism” in the Introduction and have defined its role in scene understanding.
2. We have fixed minor issues with Figure 1, Figure 2, and Appendix A.1.
3. We have moved certain results from the Discussion (Section 8) to the Results (Section 6).
4. We have added a new Figure 12 and text associated with it (Appendix A.8) to describe how pixel-level and semantic similarities between generated and original images affect metameric judgments.
5. We have added a “Limitations” section (Section 7) to describe the shortcomings of using MetamerGen to generate certain image elements (e.g. human faces, limbs, text); and its effects on the behavioral experiment if done so.
6. We have added a new ablation behavioral experiment to study the sole contributions of foveal and peripheral tokens on metameric judgments. Detailed results for this can be found in Section 6.3 and Appendix A.9.
7. We have added a new ablation computational experiment to measure the effect of blur and foveal token count on image generation quality.  Detailed results can be found in Appendix A.7.
8. We conducted a new ablation computational experiment where we assessed the use of DINOv2 and CLIP as vision encoders for MetamerGen. Detailed results can be found in Appendix A.10.

Please let us know if you have any other questions or concerns. We look forward to hearing from you and engaging in this discussion.

---

### Author Response · Authors · 2025-12-03
**Authors’ summary of review period**

We deeply thank the three reviewers, the original AC, and the replacement AC for their hard work in evaluating our manuscript. We received thoughtful and largely positive reviews from the reviewers. In an effort to increase the impact and quality of our work, we were very responsive and performed a number of additional computational and behavioral ablations to address the reviewers’ remaining suggestions. Please note that, unlike computational experiments, it is an extremely large undertaking to collect behavioral data in this short discussion period. To aid the replacement AC in their expedited consideration of our work, we summarize the reviewers’ most important suggestions and our responses here.



| **Reviewer suggestion**                                                                                                         | **Author response**                                                                                                                                                                                                                                                                                                                                       |
|---------------------------------------------------------------------------------------------------------------------------------|-----------------------------------------------------------------------------------------------------------------------------------------------------------------------------------------------------------------------------------------------------------------------------------------------------------------------------------------------------------|
| Reviewer KkGc asked whether physical distance could provide a trivial explanation of our findings.                              | We investigated and ruled out this possibility by showing that physical distance (PSNR) could not predict “same” judgments on MetamerGen’s generations, but semantic distance (DreamSim) can.                                                                                                                                                             |
| Reviewer y8G9 asked for validation of DINOv2 as a proxy for the information humans glean from scene viewing.                    | We provided this in both added theoretical justification (equating the role of the fovea and parafovea in scene viewing with the contextual information encoded by DINOv2 tokens thanks to the iBOT loss) and an empirical computational ablation (showing that DINOv2 conditioning yields more faithful and natural generations than CLIP conditioning). |
| Reviewer y8G9 asked for a systematic ablation investigating the role of foveal and peripheral conditioning in “same” judgments. | We provided this by reallocating our lab’s data collection capacity to a new behavioral ablation experiment on an exceptionally fast 5-day timeline, with the result convincingly showing that both types of conditioning contribute to “same” judgments and that peripheral information was more important than fixation information.                    |
| Reviewer KkGc requested clarifications to the text.                                                                             | We have improved organization and provided more context about metamerism.                                                                                                                                                                                                                                                                                 |
| Reviewer jZsD suggested an interesting extension to our approach, where aspect ratio and viewing area are manipulated.          | This is out of the scope of our present work, but we are considering this extension for future experiments.                                                                                                                                                                                                                                               |

\
We are grateful to the reviewers for these suggestions, which help rule out alternative explanations of our findings, and above all demonstrate the value of MetamerGen in generating stimuli that help test hypotheses about human scene understanding.

---

### Meta-Review · Area_Chair_ZrwE · 2026-01-09

**Summary:**

Reviewers were generally positive about the paper’s goal of using gaze-contingent generative modeling to study human scene understanding, but several concerns shaped the decision. The main issues were whether foveation provides unique scientific value beyond random fixations and whether the roles of foveal versus peripheral information were cleanly isolated (y8G9), whether “same” judgments could be trivially explained by low-level image similarity or partial reconstruction rather than scene understanding (KkGc), and the limited generality implied by the use of a fixed resolution and aspect ratio (jZsD).

**Reviewer Concerns:**

Most major concerns were addressed in the rebuttal.

For (y8G9), the authors added explicit behavioral ablations separating foveal-only, peripheral-only, and combined conditioning, and provided additional justification and comparisons supporting the use of DINOv2 features, largely resolving the core methodological questions.

For (KkGc), the possibility of trivial pixel-level explanations was ruled out with new analyses showing that semantic similarity, not low-level similarity, predicts “same” judgments, and background and presentation issues were corrected.

The concern raised by (jZsD) regarding fixed resolution and aspect ratio was acknowledged and scoped as a limitation, but remains an open direction for future work.

**Reviewer Scores:**

Reviewer (jZsD) would likely increase their score slightly, from 6 to around 6–7, given the clear justification of the design choices despite the remaining scope limitation.

Reviewer (KkGc) would likely maintain an accept-level score (around 8), as all major concerns were directly addressed with additional analyses and clarifications.

Reviewer (y8G9) would plausibly raise their score from 6 to around 7, since the key requested ablations and representation justifications were added, even though some broader conceptual and baseline questions remain open.

---

### Decision · Program_Chairs · 2026-01-26

Accept (Oral)